EMBO
reports

# MARCH6 and TRC8 facilitate the quality control of cytosolic and tail-anchored proteins

Sandra Stefanovic-Barrett, Anna S Dickson, Stephen P Burr, James C Williamson, Ian T Lobb, Dick JH van den Boomen, Paul J Lehner & James A Nathan*

## Abstract

**Misfolded or damaged proteins are typically targeted for destruction by proteasome-mediated degradation, but the mammalian ubiquitin machinery involved is incompletely understood. Here, using forward genetic screens in human cells, we find that the proteasome-mediated degradation of the soluble misfolded reporter, mCherry-CL1, involves two ER-resident E3 ligases, MARCH6 and TRC8. mCherry-CL1 degradation is routed via the ER membrane and dependent on the hydrophobicity of the substrate, with complete stabilisation only observed in double knockout MARCH6/TRC8 cells. To identify a more physiological correlate, we used quantitative mass spectrometry and found that TRC8 and MARCH6 depletion altered the turnover of the tail-anchored protein heme oxygenase-1 (HO-1). These E3 ligases associate with the intramembrane cleaving signal peptide peptidase (SPP) and facilitate the degradation of HO-1 following intramembrane proteolysis. Our results highlight how ER-resident ligases may target the same substrates, but work independently of each other, to optimise the protein quality control of selected soluble and tail-anchored proteins.**

**Keywords** ERAD; intramembrane proteolysis; MARCH6; protein quality control; TRC8

**Subject Categories** Post-translational Modifications, Proteolysis & Proteomics; Protein Biosynthesis & Quality Control

## Introduction

Intracellular protein quality control is essential to remove misfolded proteins that may otherwise promote aggregate formation and toxicity, leading to conditions such as Alzheimer's disease, Parkinson's disease and cystic fibrosis [1,2]. Terminally misfolded proteins typically have exposed hydrophobic domains prone to aggregation [3] and can be ubiquitinated and selectively targeted for proteasomal degradation. In yeast, these proteins are ubiquitinated by quality control E3 ligases, including the cytosolic Ubr1p and San1p ligases [4–6] and the endoplasmic reticulum (ER)-resident ligase Doa10p [7–9]. The mammalian ubiquitin machinery required to remove aberrantly folded proteins is less clearly defined. The increased diversity and complexity seen in mammalian enzymes involved in protein quality control are in part driven by an expansion of E2 and E3 ligases [10], highlighting the need to elucidate their detailed functions.

ER-associated degradation (ERAD) provides the major mechanism for protein quality control at the ER membrane, facilitating the dislocation of proteins from the ER for degradation by the proteasome within the cytosol. In yeast, three E3 ligases promote ERAD: Hrd1p and Doa10p recognise their substrates at the ER [11], while the Asi E3 ligase complex facilitates ERAD at the inner nuclear membrane [12,13]. In mammalian cells, orthologues of Hrd1p (HRD1) and Doa10p (MARCH6, also known as TEB4) support ERAD [14–17], but there has also been a marked expansion of ER-resident E3 ligases. Some of these are homologous to Hrd1p (e.g. gp78) [18], while others do not resemble the yeast ligases, such as the RING-H2 containing TRC8 [19] and TMEM129 [20,21], both implicated in the human cytomegalovirus-mediated ERAD of MHC class I molecules.

Several reporter systems for investigating protein quality control have been developed, including the generation of artificial substrates with exposed hydrophobic domains typical of misfolded proteins [22]. The CL1 degron was originally identified in a *Saccharomyces cerevisiae* (*S. cerevisiae*) screen for proteasome substrates as an unstable C-terminal extension that promotes rapid degradation [7,23], and has been widely used to report on proteasome activity in the context of neurological diseases associated with cytosolic protein aggregation [24,25]. In yeast, attachment of CL1 to cytosolic proteins (e.g. Ura3p) leads to their degradation via the ERAD degradation machinery, including the ER-resident E3 ligase Doa10p [23]. In mammalian cells, fusion of this 16 amino acid amphipathic helix to usually stable proteins also leads to their proteasomal destruction [22,26], but the molecular machinery required for degradation is not known.

Here, we use the CL1 degron as a model substrate in unbiased mutagenesis screens to identify the ubiquitin machinery required for degrading misfolded soluble proteins in human cells. We show that two ER-resident ubiquitin ligases, TRC8 and MARCH6, facilitate the proteasomal degradation of the soluble fluorescent CL1 substrate (mCherry-CL1) at the cytosolic face of the ER membrane.

Department of Medicine, Cambridge Institute for Medical Research, University of Cambridge, Cambridge, UK
*Corresponding author. Tel: +44 1223 748483; E-mail: jan33@cam.ac.uk

To establish the biological significance of this pathway, we use a quantitative proteomic approach to identify potential substrates of these ER-resident ligases. We find that selected tail-anchored proteins that undergo intramembrane proteolysis by signal peptide peptidase (SPP) [27] share some properties with the mCherry-CL1 degron, in that their degradation can be mediated by both TRC8 and MARCH6.

## Results

### A near-haploid genetic screen identifies TRC8 and MARCH6 as ubiquitin E3 ligases involved in degradation of the CL1 degron

We established a human mutagenesis screen in near-haploid KBM7 cells (karyotype 25, XY, +8, Ph+) [28] to identify genes required for protein quality control, using a fluorescent CL1 degron as an unstable hydrophobic proteasome substrate. This screening methodology provides a powerful unbiased forward genetic approach, which has led to the identification of novel pathogen restriction factors, epigenetic regulators, ERAD components and genes involved in regulation of the hypoxia response [28–33]. We designed our fluorescent reporter to encode the CL1 amphipathic sequence as a C-terminal extension of mCherry (Fig 1A and B). KBM7 and HeLa cells stably expressing this reporter showed low mCherry-CL1 fluorescence and protein levels, which increased following incubation with the proteasome inhibitor, bortezomib (Velcade), consistent with constitutive proteasome-mediated degradation of the reporter (Fig 1C and D). Confocal immunofluorescence demonstrated that mCherry-CL1 was diffusely distributed within the cytosol following proteasome inhibition (Fig 1E), as observed by others for a GFP-CL1 fusion protein in human cells [22,26].

The genetic screen was performed by insertional mutagenesis of an mCherry-CL1 KBM7 clone with a gene-trapping retrovirus, and selective enrichment of high mCherry fluorescence (mCherry[HIGH]) by two rounds of fluorescence-activated cell sorting (FACS) (Fig 1F). Gene-trapping insertions were identified by Illumina sequencing in the mCherry[HIGH] population and compared to a control library of gene-trap mutagenised cells that had not been phenotypically selected (Fig 1G and Dataset EV1). Five genes were highly enriched for trapping insertions (Fig 1G and H), including two ubiquitin E3 ligases, TRC8 and MARCH6; the ubiquitin E2 conjugating enzyme, UBE2G2; and the ubiquitin-binding protein,

AUP1. The fifth gene, SPEN, is frequently enriched by this type of screening approach and is not specific to the mCherry-CL1 reporter.

UBE2G2 and AUP1 (Fig 1I) are orthologues of yeast Ubc7p and Cue1p and form a complex that promotes ubiquitination at the ER membrane through an E2-binding domain on AUP1 (G2BR on AUP1, U7BR on Cue1p) [8,34,35]. Interestingly, Ubc7p and Cue1p were shown to be required for the degradation of Ura3p-CL1 in yeast [23]. However, in contrast to the studies in yeast where the MARCH6 homologue, Doa10p, is sufficient for degradation of CL1 fusion proteins, the human genetic screen identified both TRC8 and MARCH6, predicting a non-redundant role for these two ER-resident E3 ligases (Fig 1I) in the degradation of mCherry-CL1 in human cells.

### Combined depletion of both MARCH6 and TRC8 stabilises mCherry-CL1 to levels observed with proteasome inhibition

We validated a role for MARCH6, TRC8, UBE2G2 and AUP1 in the degradation of mCherry-CL1 using CRISPR/Cas9 depletion. HeLa cells were transiently transfected with Cas9 and sgRNA to all four genes and mCherry-CL1 fluorescence measured by flow cytometry on day 7. Depletion of each gene increased levels of mCherry-CL1 fluorescence but never reached the control levels observed with proteasome inhibition (Fig 2A). These findings were specific to ER ligases identified in the screen, as depletion of another ER ligase, gp78, had no effect on mCherry-CL1 levels (Appendix Fig S1A and B).

Knockout (KO) clones isolated from the CRISPR targeted populations confirmed that specific loss of MARCH6, TRC8 E3, UBE2G2 or AUP1 increased mCherry-CL1 fluorescence and protein levels (Fig 2B–D). Due to the lack of a suitable MARCH6 antibody, a functional clonal depletion of MARCH6 was validated through the stabilisation of squalene monooxygenase (SQLE), a previously reported MARCH6 substrate [16,36], and the presence of indel formation within the targeted MARCH6 locus (Fig 2D, Appendix Fig S2). Reconstitution of the UBE2G2, TRC8 and MARCH6 null clones with the respective wild-type cDNA restored mCherry-CL1 degradation, while catalytically inactive forms of UBE2G2 (UBE2G2 C89A), TRC8 (TRC8 ΔRING) or MARCH6 (MARCH6 C9A) had no effect on mCherry-CL1 levels (Appendix Fig S3A–F), further confirming the specificity of the gene KOs. Moreover, overexpression of the catalytically inactive enzymes (UBE2G2 C89A, TRC8 ΔRING or MARCH6

---

**Figure 1. A near-haploid genetic screen identifies genes required for cytosolic protein quality control in human cells.**

A  Schematic of the mCherry-CL1 reporter.

B  Diagram of the 16 amino acid CL1 amphipathic helix using a helical wheel prediction (http://rzlab.ucr.edu/scripts/wheel/wheel.cgi).

C, D  Clonal populations of KBM7 (left) or HeLa cells (right) expressing the mCherry-CL1 reporter were incubated with 40 nM (KBM7) or 100 nM (HeLa) bortezomib (Btz) for 6 h and mCherry levels measured by flow cytometry (C) or immunoblot (D). *Presumed degradation product of mCherry-CL1, present to a variable extent when degradation is impaired. β-actin served as a loading control.

E  Confocal microscopy of HeLa mCherry-CL1 cells treated with or without 100 nM bortezomib for 6 h. Scale bar, 10 μm.

F  KBM7 forward genetic screen with a mCherry-CL1-expressing clone. The cells were mutagenised with the Z-loxP-GFP gene-trap retrovirus, enriched for mCherry[HIGH] cells by FACS, and insertion sites identified by MiSeq Illumina sequencing.

G  Bubble plot of enriched genes in the mCherry[HIGH] population compared to unsorted mutagenised control KBM7 cells. Bubble size is proportional to the number of independent inactivating gene-trap integrations identified (shown in brackets). Genes that were significantly enriched (> −log(p)5) are shown (SPEN is found to be frequently enriched for gene-trap insertions in these types of screens and was not taken forward for further validation).

H  Location of the enriched gene-trap insertions in AUP1, UBE2G2, TRC8 and MARCH6 genes (red, sense insertion; blue, antisense insertion). The predominance of insertions in the correct orientation at the start of the gene indicates enrichment for gene-trapping mutations.

I  Schematic of AUP1, UBE2G2, TRC8 and MARCH6 in the ER membrane. The position of the E3 ligase RING domain is shown. SSC = side scatter.

    

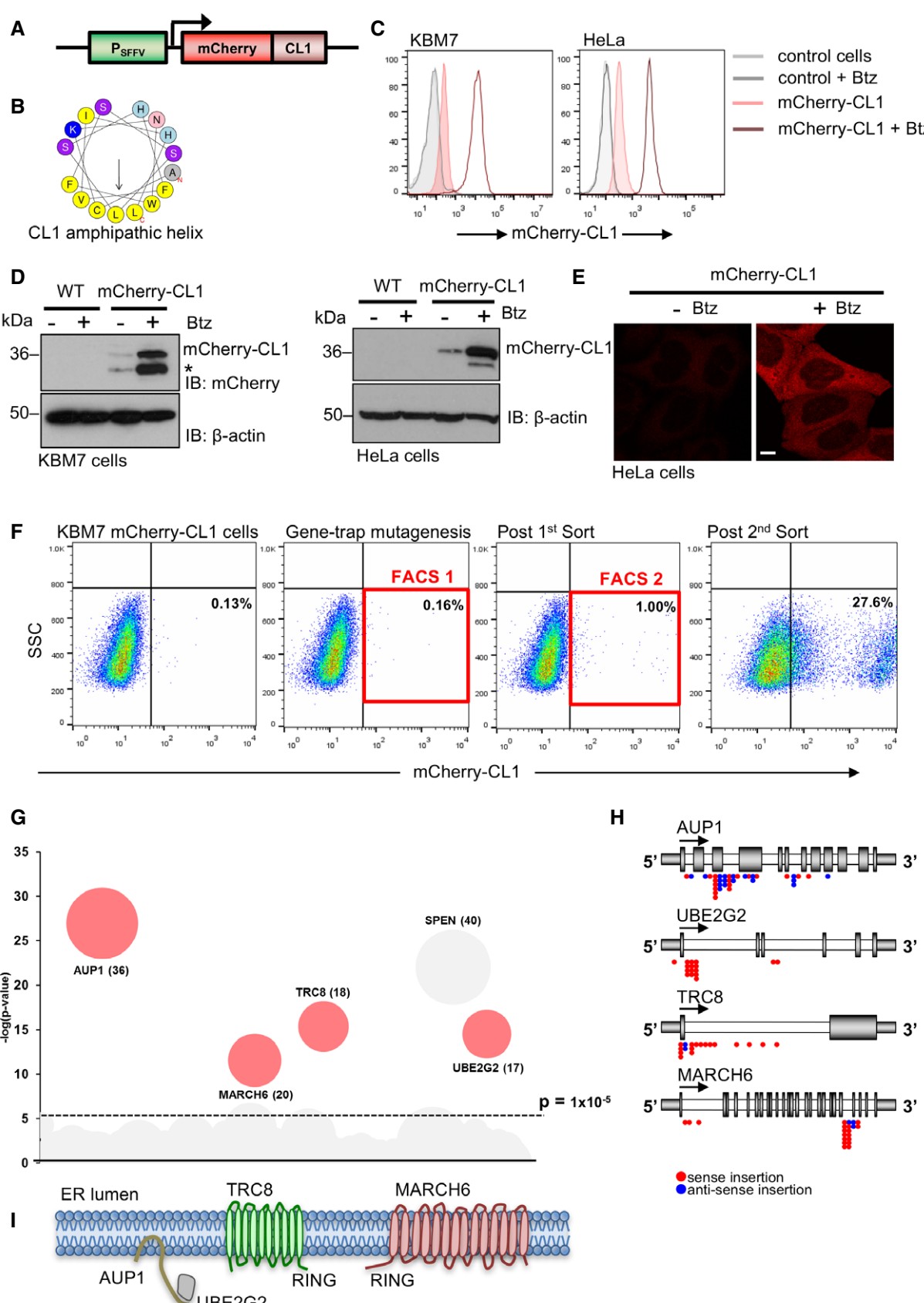

**Figure 1.**

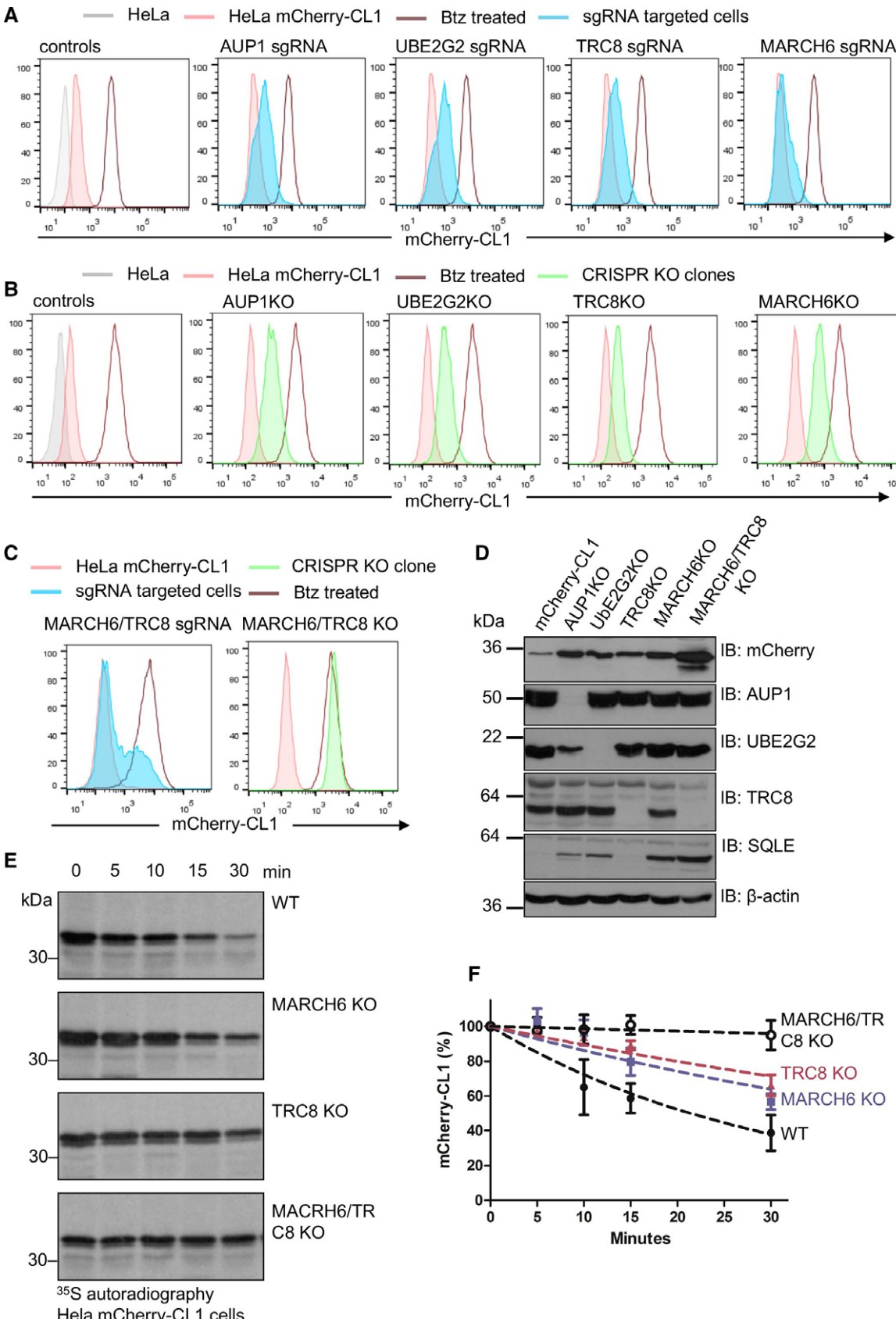

**Figure 2.**

**Figure 2.   Depletion of both MARCH6 and TRC8 is required for the stabilisation of mCherry-CL1.**

A    HeLa mCherry-CL1 cells were transiently transfected with Cas9 and sgRNA targeting AUP, UBE2G2, TRC8 or MARCH6, and mCherry fluorescence measured by flow cytometry after 7 days (blue). 20 nM bortezomib 16 h was used as a control for mCherry-CL1 stabilisation (brown line).

B    Representative mCherry-CL1 levels (green) in isolated KO clones for AUP1, UBE2G2, TRC8 and MARCH6 from the sgRNA targeted cells (A).

C    Mixed populations (shaded blue) and clonal population (shaded green) of combined TRC8/MARCH6 KO cells. 20 nM bortezomib 16 h was used as a control for mCherry-CL1 stabilisation (brown line).

D    Representative immunoblot for mCherry levels in the AUP1, UBE2G2, MARCH6 and TRC8 in the HeLa mCherry-CL1 KO clones. Antibodies for AUP1, UBE2G2 and TRC8 confirmed loss of protein in the relevant clone. SQLE levels are shown in support of MARCH6 deficiency. β-actin served as a loading control.

E, F    [35S]methionine/cysteine-radiolabelling of mCherry-CL1 cells in wild-type (WT), MARCH6, TRC8 or combined MARCH6/TRC8 null cells. Cells were pulse-radiolabelled for 10 min, and mCherry immunoprecipitated from detergent lysates at the times indicated. mCherry-CL1 levels were measured by autoradiography (E) and quantification (using ImageJ and GraphPad Prism) of mCherry-CL1 degradation is shown (F). Mean ± SEM, n = 3.

C9A) in mCherry-CL1 HeLa cells stabilised the fluorescent protein (Appendix Fig S3B, D and F), demonstrating a dominant negative effect of these mutants. Reconstitution of the AUP1 null clones with wild-type AUP1 did not restore mCherry-CL1 degradation, principally due to the toxicity of ectopic AUP1. However, it was likely that AUP1 loss resulted in destabilisation of UBE2G2 (Fig 2D, Appendix Fig S3G), similar to the *S. cerevisiae* orthologues, Cue1p and Ubc7p [37].

The incomplete rescue of mCherry-CL1 following TRC8 or MARCH6 depletion (Fig 2A) was consistent with a requirement for both ligases. To explore this further, we measured the rate of mCherry-CL1 degradation by [35S]methionine/cysteine-radiolabelling in MARCH6 or TRC8 null cells (Fig 2E and F). mCherry-CL1 was rapidly degraded in wild-type HeLa cells with a half-life of 21 min (± 5 min) (Fig 2E and F). mCherry-CL1 degradation was also decreased in both the MARCH6 null and TRC8 null cells, but neither completely stabilised the protein (MARCH6 KO half-life 42 min ± 18, TRC8 KO half-life 62 min ± 20; Fig 2E and F). Only when both ligases were depleted did we observe an increase in mCherry-CL1 to the equivalent level seen following proteasome inhibition (Fig 2C and D) and stabilisation of the protein (Fig 2E and F).

Both the MARCH6 and TRC8 catalytically inactive mutants failed to restore mCherry-CL1 degradation in the respective null clones (Appendix Fig S3), demonstrating a requirement for their ligase activity. To confirm that the ligases altered CL1 ubiquitination, we measured the accumulation of polyubiquitinated mCherry-CL1 following proteasome inhibition (MG132, 50 μM for 2 h) in wild-type or combined MARCH6/TRC8 null cells (Appendix Fig S4A). There was a marked reduction in polyubiquitinated mCherry-CL1 in the MARCH6-/TRC8-deficient cells compared to the HeLa controls (Appendix Fig S4A). The low level of residual ubiquitination in the combined null cells may reflect MARCH6- or TRC8-independent ubiquitination, but this did not seem to significantly contribute to mCherry-CL1 degradation (Fig 2F).

To determine which polyubiquitin linkages were involved in CL1 degradation, we used wild-type ubiquitin and ubiquitin lysine (K) mutants, encoding a GFP fusion that is co-translationally cleaved, similar to endogenous ubiquitin processing [38,39]. Therefore, GFP levels provide a quantitative marker of ubiquitin expression and allow gating of a GFP[HIGH] population by flow cytometry, so that only those cells in which mutant ubiquitin outcompetes endogenous ubiquitin are analysed (Appendix Fig S4B and C). mCherry-CL1 levels increased in cells overexpressing a ubiquitin mutant incapable of forming K48 linkages (Ub-K48R), but showed no change in cells expressing Ub-K63R or Ub-K11R mutants (Appendix Fig S4C). Moreover, mCherry-CL1 levels were rescued using a K48-only ubiquitin construct compared to a Ub-K0 mutant (all lysines mutated to arginines; Appendix Fig S4B). Similar findings were observed when these ubiquitin mutants were expressed in MARCH6 or TRC8 null clones (Appendix Fig S4B and C), consistent with both ligases forming K48-ubiquitin linkages.

**Degradation of mCherry-CL1 by MARCH6 or TRC8 is dependent on the hydrophobicity and membrane association of its amphipathic helix**

Our findings suggested that ubiquitination and degradation of mCherry-CL1 were initiated at the ER membrane, but the CL1 degron is typically used as a soluble cytosolic proteasome reporter in human cells. Confocal microscopy confirmed that mCherry-CL1 was diffusely distributed throughout the cell following proteasome inhibition (Figs 1E and 3A), consistent with prior reports [22]. However, some ER co-localisation was observed (Fig 3A) and we therefore probed subcellular fractionations to determine whether mCherry-CL1 associated with membranous compartments following proteasome inhibition. mCherry-CL1 was predominantly detected in the soluble (cytosolic) fraction at steady state but following proteasome inhibition, the degron was stabilised and equally distributed in the cytosol and membrane

**Figure 3.   MARCH6 and TRC8 degrade membrane-associated mCherry-CL1.**

A    Confocal microscopy of mCherry-CL1 HeLa cells treated with 20 nM bortezomib for 16 h. A KDEL antibody was used as an ER marker. Scale bar, 10 μm.

B, C    Membrane fractionation studies for mCherry-CL1 levels in HeLa mCherry-CL1 cells with or without proteasome inhibition (20 nM bortezomib 16 h) (B), or in AUP1, UBE2G2, MARCH6 or TRC8 KO HeLa mCherry-CL1 clones (C). Briefly, cells were lysed by ball-bearing homogenisation in a sucrose buffer, and the supernatant was ultracentrifuged at 50,000 rpm for 1 h to obtain the cytosol (C) and membrane (M) fractions. Tubulin and calnexin were used as control for the cytosolic and membrane fractions, respectively.

D–G    Ectopic expression of TRC8 (D, E) or MARCH6 (F, G) in combined MARCH6/TRC8 null cells. Catalytically inactive mutants (MARCH6 C9A-HA and TRC8ΔRING-HA) were also overexpressed in the MARCH6/TRC8 null cells. mCherry-CL1 levels were measured by flow cytometry and gated for HA-positive cells (black line) (D, F). Basal mCherry-CL1 levels in the parent reporter cells (red) and combined MARCH6/TRC8 null cells (green) are shown. HA-tagged overexpressed ligases were also visualised by immunoblot (E, G).

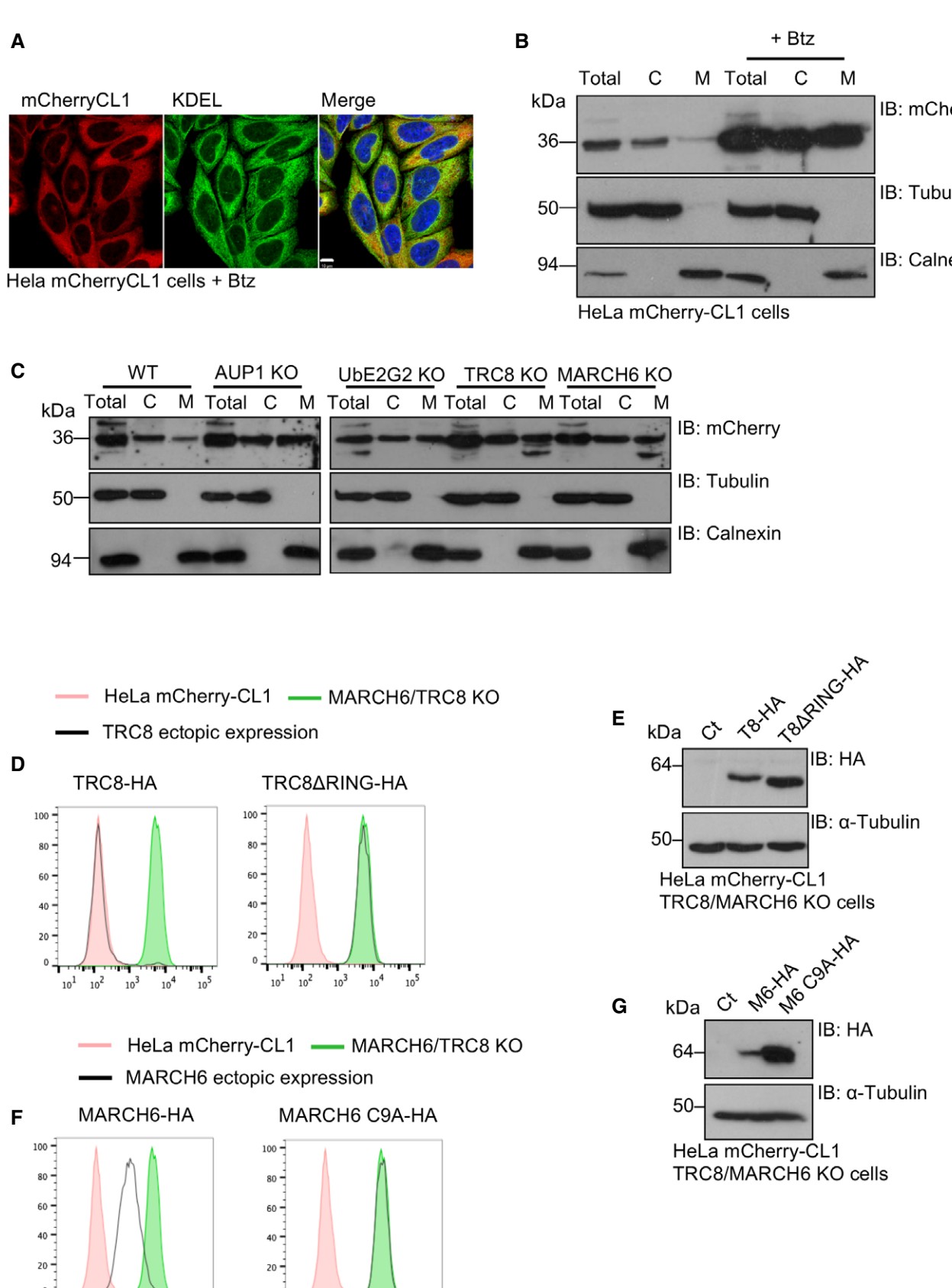

Figure 3.

fractions (Fig 3B). Similar findings were observed in HeLa mCherry-CL1 AUP1, UBE2G2, TRC8 and MARCH6 null cells (Fig 3C), with all four KO lines showing an increase in the membrane fraction of mCherry-CL1 compared to the wild-type HeLa mCherry-CL1 lysates (Fig 3C).

Proteins with exposed hydrophobic domains are often bound by chaperones to assist folding and promote degradation. Chaperones are also required to deliver transmembrane proteins to the ER and facilitate their insertion. The chaperone Bag6 (also known as BAT3 or Scythe), which forms part of the guided entry of tail-anchored proteins (GET)/transmembrane recognition complex (TRC), binds CL1 and is implicated in the recognition and subsequent ubiquitination of this degron [40]. Neither Bag6 nor other chaperones were identified in the genetic screen, but these screens are unlikely to reach saturation. We therefore depleted cells of Bag6 and visualised mCherry levels by flow cytometry (Appendix Fig S5A and B). Bag6 depletion destabilised the GET/TRC pathway, as evidenced by decreased Ubl4A levels [41] (Appendix Fig S5C), but did not affect mCherry-CL1 levels (Appendix Fig S5A and B). These data are consistent with proteasome-mediated degradation of mCherry-CL1, being dependent on its ER membrane association but independent of Bag6.

The partial stabilisation of mCherry-CL1 at the membrane by MARCH6 or TRC8 depletion suggested they have overlapping functions but act independently. Differences in mCherry-CL1 stabilisation were unlikely to be due to the type of ubiquitin chain formed, as both ligases showed a requirement for forming K48-ubiquitin linkages (Appendix Fig S4B and C) However, consistent with their overlapping functions, ectopic expression of either MARCH6 and TRC8 decreased the fluorescent reporter levels in the combined MARCH6-/TRC8-deficient cells (Fig 3D–G). Indeed, overexpression of TRC8 completely restored mCherry-CL1 to basal levels (Fig 3D and E) (it was possible that MARCH6 only partially restored mCherry-CL1 degradation due to its rapid autoubiquitination and degradation [42]).

We next explored whether the hydrophobicity of the CL1 amphipathic helix could explain the overlapping functions or specificity of the ligases. Several mutations were introduced by substituting alanine for hydrophobic residues within the CL1 amphipathic sequence (CL1 2A, CL1 4A and CL1 6A) to generate mCherry-CL1 constructs with varying degrees of hydrophobicity (Fig 4A). Stable expression of these fluorescent CL1 reporters in HeLa cells showed that their protein levels increased as hydrophobicity decreased, with mutations of all the bulky hydrophobic residues (CL1 6A) resulting in complete stabilisation of the reporter (Fig 4B, bottom panel). Moreover, as the hydrophobicity of the CL1 reporters decreased, their membrane association also decreased (Fig 4C). Indeed, once the CL1 protein

was fully soluble and not hydrophobic (CL1 6A), it no longer associated with membranes and was not degraded (Fig 4C, bottom panel). Thus, CL1 hydrophobicity correlated with membrane association and proteasome-mediated degradation.

To determine whether the CL1 mutants with decreased hydrophobicity were still degraded by MARCH6 or TRC8, we measured their mCherry levels in mixed CRISPR KO populations of each ligase, or following combined MARCH6/TRC8 depletion (Fig 4B). The unmodified mCherry-CL1 increased following TRC8 or MARCH depletion but only stabilised to the level of proteasome inhibition following combined TRC8/MARCH6 loss (Fig 4B, top panel), consistent with our prior results. However, mutations of the bulky hydrophobic residues altered the ligase specificity for CL1 degron, such that both the CL1 2A and 4A mutations were degraded solely in a TRC8-dependent manner (Fig 4B, middle panels). Moreover, depletion of TRC8 was sufficient to stabilise mCherry-CL1 2A and 4A to the same level as proteasome inhibition, with no additional change in mCherry levels observed following combined MARCH6/TRC8 loss (Fig 4B, middle panels). TRC8 or MARCH6 depletion had no effect on the levels of mCherry-CL1 6A (Fig 4B, bottom panel), but this was expected as the CL1 reporter no longer associated with membranes (Fig 4C, bottom panel).

Lastly, we mutated the single lysine residue within the CL1 amphipathic helix (CL1 K3A) to determine whether this was required for ubiquitination or whether the charged residue was important for degradation (Fig 4D). The mCherry-CL1 K3A reporter was still rapidly degraded in HeLa cells, indicating that the lysine was not required for degradation (Fig 4D). Moreover, depletion of both MARCH6 and TRC8 was required for full stabilisation of mCherry-CL1 K3A, similar to the unmodified CL1 helix (Fig 4B, top panel and 4D).

Thus, while both MARCH6 and TRC8 recognise the hydrophobic membrane-associated region of the CL1 degron, the degree of hydrophobicity is important, with MARCH6 requiring the longer hydrophobic region of the CL1 sequence, compared to TRC8, which can degrade the less hydrophobic CL1 mutant reporters.

## AUP1, UBE2G2 and TRC8 facilitate the degradation of the mCherry-CL1 through a shared pathway

The incomplete rescue of mCherry-CL1 degradation in the AUP1 or UBE2G2 KO clones implied that AUP1 and UBE2G2 functioned with only one of the two ligases identified in the genetic screen. To determine the correct E2/E3 pairing, we depleted cells of AUP1 or UBE2G2 in combination with TRC8 or MARCH6 and determined which double gene KO provided a complete recovery of mCherry-CL1 fluorescence (Fig 5A and B). No combination of sgRNA targeting TRC8 with AUP1 or UBE2G2 led to a complete

**Figure 4. CL1 hydrophobicity determines the specificity for recognition by MARCH6 or TRC8.**

A, B  HeLa cells stably expressing wild-type mCherry-CL1 or mCherry-CL1 mutants with alanine (red) mutations of the bulky hydrophobic residues (blue) (A) were transiently transfected with Cas9 and sgRNA targeting MARCH6, TRC8 or both (B). Hydrophobicity (arbitrary units) generated using "helical wheel projection" created by Don Armstrong and Raphael Zidovetzki (Version Id: wheel.pl, v 1.4 2009-10-20 21:23:36 don Exp.). Helical wheel generated using HeliQuest [75] (A). mCherry-CL1 wild-type and mutant levels (red) were measured following sgRNA transfection for the ligases (blue) or after treatment with bortezomib (Btz) (brown line) (B).

C  Membrane fractionation studies for mCherry-CL1 wild-type or CL1 mutants (CL1 4A, 2A and 6A) as previously described.

D  HeLa cells stably expressing the mCherry-CL1 K3A mutant were transiently transfected with Cas9 and sgRNA targeting MARCH6, TRC8 or both as described.

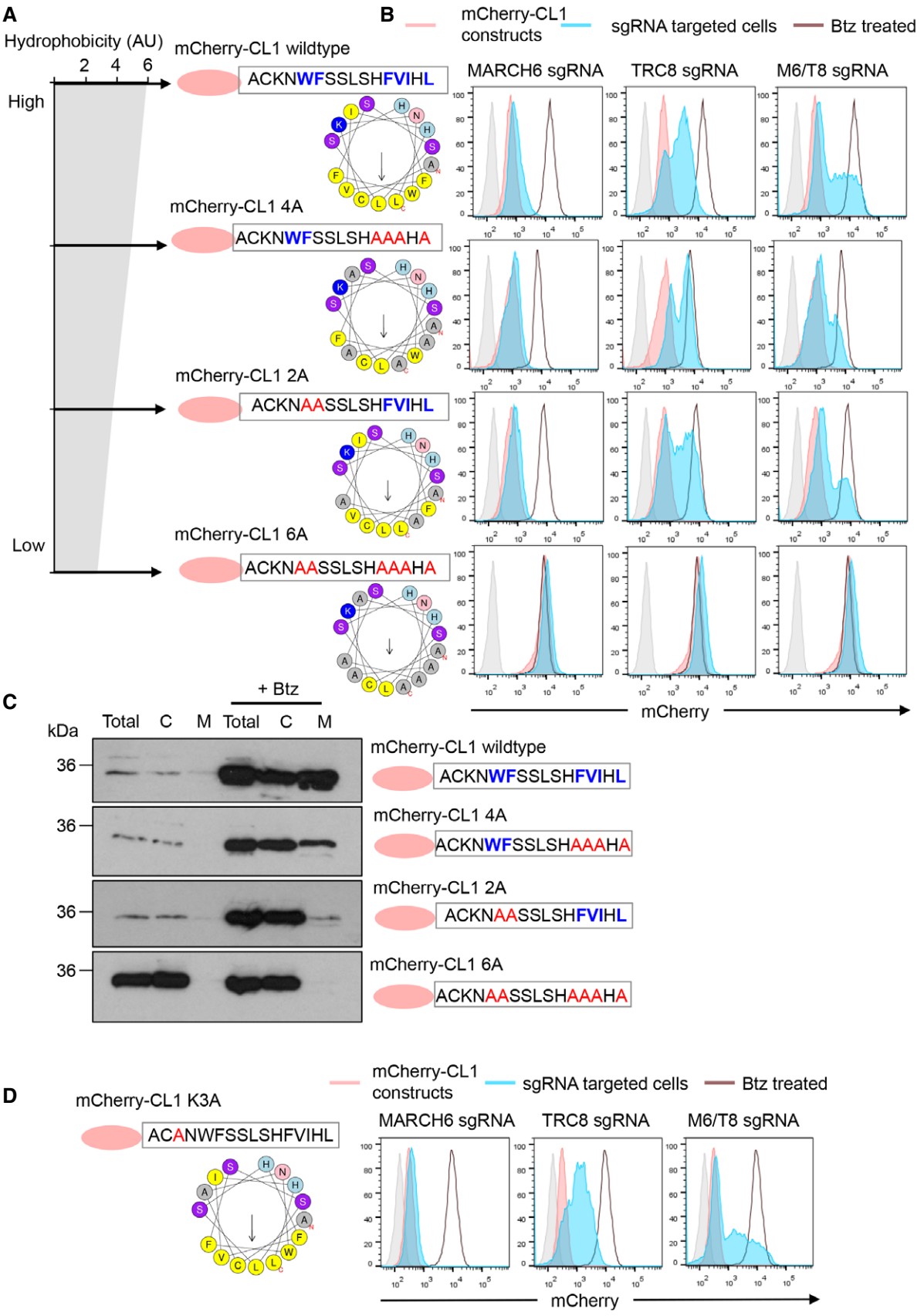

**Figure 4.**

mCherry-CL1 recovery, as fluorescence was never increased further than observed for the single-gene knockouts (Fig 5A). However, depletion of MARCH6 in combination with AUP1 or UBE2G2 rescued mCherry-CL1 to levels seen with proteasome inhibition or combined MARCH6/TRC8 depletion (Fig 5B). These genetic data suggest that UBE2G2 and AUP1 functioned within the same pathway as TRC8.

To further confirm these findings (Fig 5C), we depleted UBE2G2 or AUP1 in a TRC8 null clone. As expected, AUP1 and UBE2G2 depletion did not alter mCherry-CL1 levels in the TRC8 KO cells (Fig 5C). In contrast, when MARCH6 was depleted from the mCherry-CL1 TRC8 KO cells, full stabilisation of mCherry-CL1 was observed (Fig 5C). Similarly, depletion of UBE2G2 or TRC8 did not alter mCherry-CL1 levels in AUP1 cells (Fig 5D). Complete stabilisation of mCherry-CL1 was only achieved upon depletion of MARCH6 in the AUP1 null cells (Fig 5D). Together, these genetic studies show that TRC8, UBE2G2 and AUP1 function within the same pathway, independently of MARCH6.

### A CRISPR/Cas9 forward genetic screen identifies UBE2J2 as an E2 conjugating enzyme for MARCH6

These data implied that the KBM7 genetic screen was not saturating and a candidate E2 conjugating enzyme for MARCH6 remained to be identified. We therefore adopted an orthogonal strategy and established a CRISPR/Cas9 genetic screen to identify the cognate E2 enzymes for MARCH6 ubiquitination of mCherry-CL1. We screened on the UBE2G2 null cells expressing the mCherry-CL1 reporter, as this would favour the detection of E2 enzymes involved in MARCH6-mediated degradation (Fig 6A). Briefly, a HeLa UBE2G2 null clone expressing Cas9 and the mCherry-CL1 degron was transduced with a genome-wide sgRNA library (Brunello library: 76,441 sgRNA) [43] and enriched for high mCherry fluorescence (mCherry[HIGH]) by two rounds of flow cytometry sorting (Fig 6A). SgRNA insertions were identified by Illumina HiSeq, and E2 enzymes enriched for indels were identified by comparing the number of sgRNA sequences targeting E2 enzymes in the mCherry[HIGH] population to unsorted HeLa mCherry-CL1 UBE2G2 null cells transduced with the sgRNA library (Fig 6B and Dataset EV2). One E2 enzyme, UBE2J2, was significantly enriched for sgRNA targeting compared to the unselected sgRNA transduced populations ($P < 10^{-5}$) and was particularly relevant, as it localises to the ER membrane and is involved in ERAD [20,44]. Two additional E2 enzymes, UBE2D3 (also known as UbcH5c) and UBE2K (also known as HIP2), were enriched at borderline significance (Fig 6B).

We validated the E2 enzymes identified in the screen by transient sgRNA targeting of UBE2J2, UBE2D3 or UBE2K in mCherry-CL1 HeLa cells (Fig 6C and D). UBE2J2 KO populations were confirmed by immunoblot (Fig 6E) and increased mCherry-CL1 levels to intermediate levels, while depletion of UBE2D3 or UBE2K alone had minimal effects (Fig 6C and D). The combined depletion of UBE2J2 and UBE2D3 or UBE2G2 and UBE2D3 increased mCherry-CL1 levels further, but complete stabilisation (to the level of proteasome inhibition) of the CL1 degron was only achieved following the combined depletion of all three UBE2J2, UBE2G2 and UBE2D3 enzymes (Fig 6D). Combined depletion of UBE2J2 with UBE2K had no additive effect. These findings implicate UBE2J2 and UBE2D3 as the two E2 enzymes recruited by MARCH6.

To further clarify that UBE2J2 worked with MARCH6, we examined the effect of UBE2J2 depletion on the levels of the MARCH6 substrate, SQLE. Depletion of UBE2J2 increased SQLE levels similar to those observed following proteasome inhibition or MARCH6 depletion (Fig 6E), while in control HeLa cells, depletion of UBE2G2 or UBE2D3 had little effect on SQLE (Fig 6F). Moreover, we observed that the catalytically inactive form of MARCH6 associated with FLAG tagged UBE2J2 (Fig 6G). These findings are consistent with UBE2J2 acting as the predominant E2 conjugating enzyme for MARCH6 in the context of SQLE ubiquitination.

### MARCH6 and TRC8 facilitate the degradation of selected ER proteins following SPP-mediated intramembrane proteolysis

The degradation of the CL1 degron by MARCH6 and TRC8 suggested overlapping functions for these ligases in protein quality control. To determine whether any endogenous proteins were dependent on both MARCH6 and TRC8 for stability, we used tandem-mass tagging (TMT) quantitative mass spectrometry [45]. Whole-cell proteomes from HeLa mCherry-CL1 cells and the combined MARCH6/TRC8 KO clones were compared using TMT-based mass spectrometry (LC-MS$^3$) (Fig 7A and Dataset EV3). MARCH6 and TRC8 KO lysates were also analysed to identify proteins that accumulated following depletion of a single ligase (Fig 7A and Dataset EV3). While several proteins were more abundant in the individual E3 ligase-deficient cells, only five proteins accumulated solely in the combined MARCH6 and TRC8 null cells with high confidence: sterol regulatory element-binding protein (SREBP) 1 and 2, acyl-CoA synthetase 1 (ACLS1), ORMDL2 (also known as adolpin-2) and HO-1.

We were particularly interested in the tail-anchored protein HO-1, as our previous study implicated a critical role of SPP-mediated intramembrane proteolysis, together with TRC8 ubiquitination, in

---

**Figure 5.  AUP, UBE2G2 and TRC8 facilitate the degradation of mCherry-CL1 through a shared pathway.**

A    HeLa mCherry-CL1 cells (shaded red) were transiently transfected with Cas9 and combinations sgRNA targeting AUP1, UBE2G2 or TRC8, generating mixed KO populations of single genes or combinations (AUP1/UBE2G2, TRC8/AUP1 or TRC8/UBE2G2, shown in shaded blue). mCherry levels were measured by flow cytometry, and bortezomib (Btz) (20 nM 16 h) was used as control for mCherry-CL1 stabilisation (brown line).

B    HeLa mCherry-CL1 cells were transiently transfected with Cas9 and sgRNA targeting MARCH6, or in combination with AUP1 or UBE2G2, generating mixed KO populations. mCherry levels were measured by flow cytometry, and bortezomib (20 nM 16 h) was used as control for mCherry-CL1 stabilisation.

C    An mCherry-CL1 TRC8 null clone (shaded green) was transiently transfected with Cas9 and sgRNA targeting AUP1, UBE2G2, MARCH6 or the empty sgRNA vector (EV) as described (blue line). Basal mCherry-CL1 levels are also shown (shaded red).

D    mCherry-CL1 AUP1 null cells (shaded green) were transiently transfected with sgRNA to UBE2G2, TRC8 or MARCH6 (blue line), and mCherry levels were measured by flow cytometry. Bortezomib (20 nM 16 h) was used as control for mCherry-CL1 stabilisation (brown line).

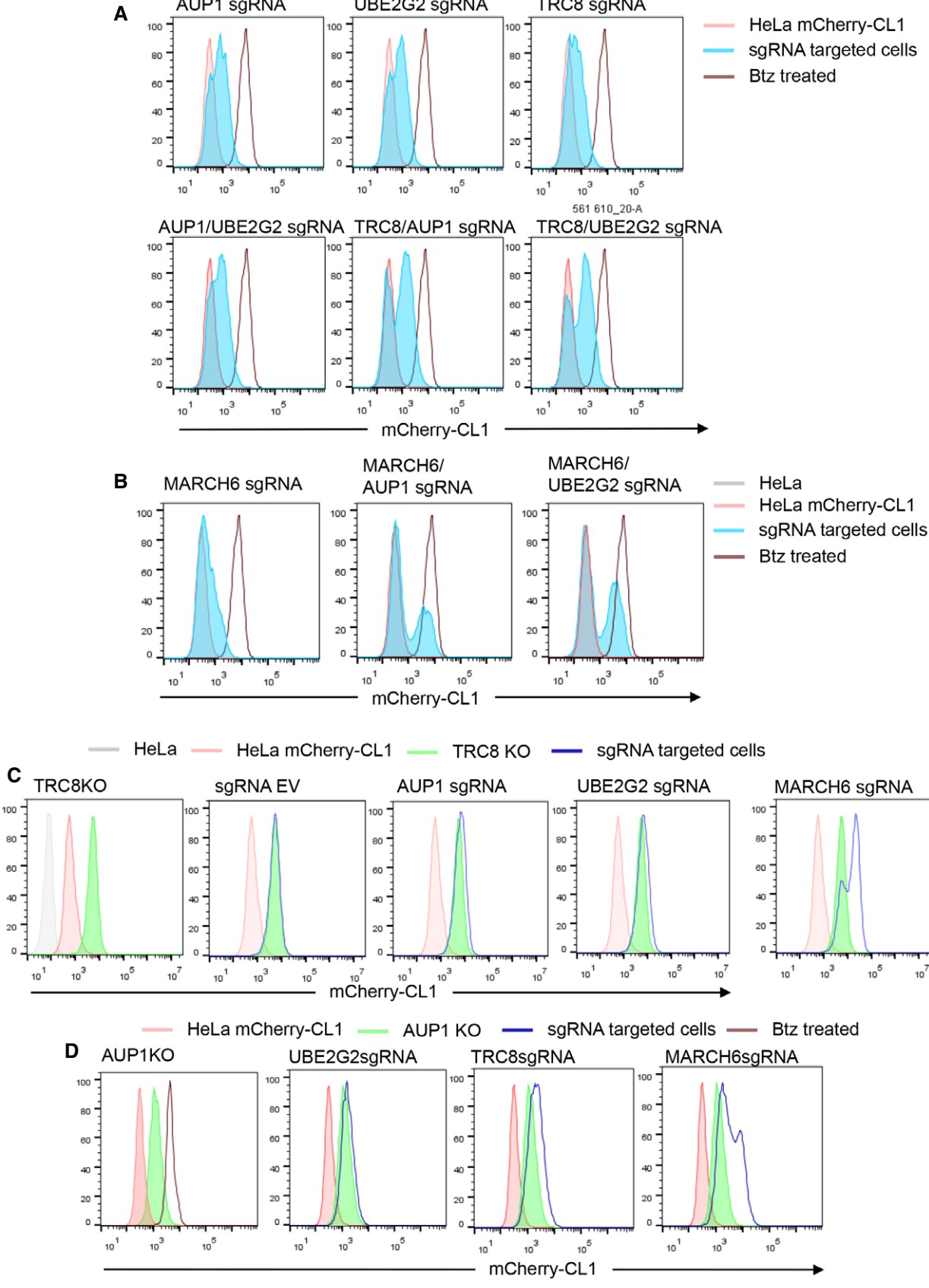

**Figure 5.**

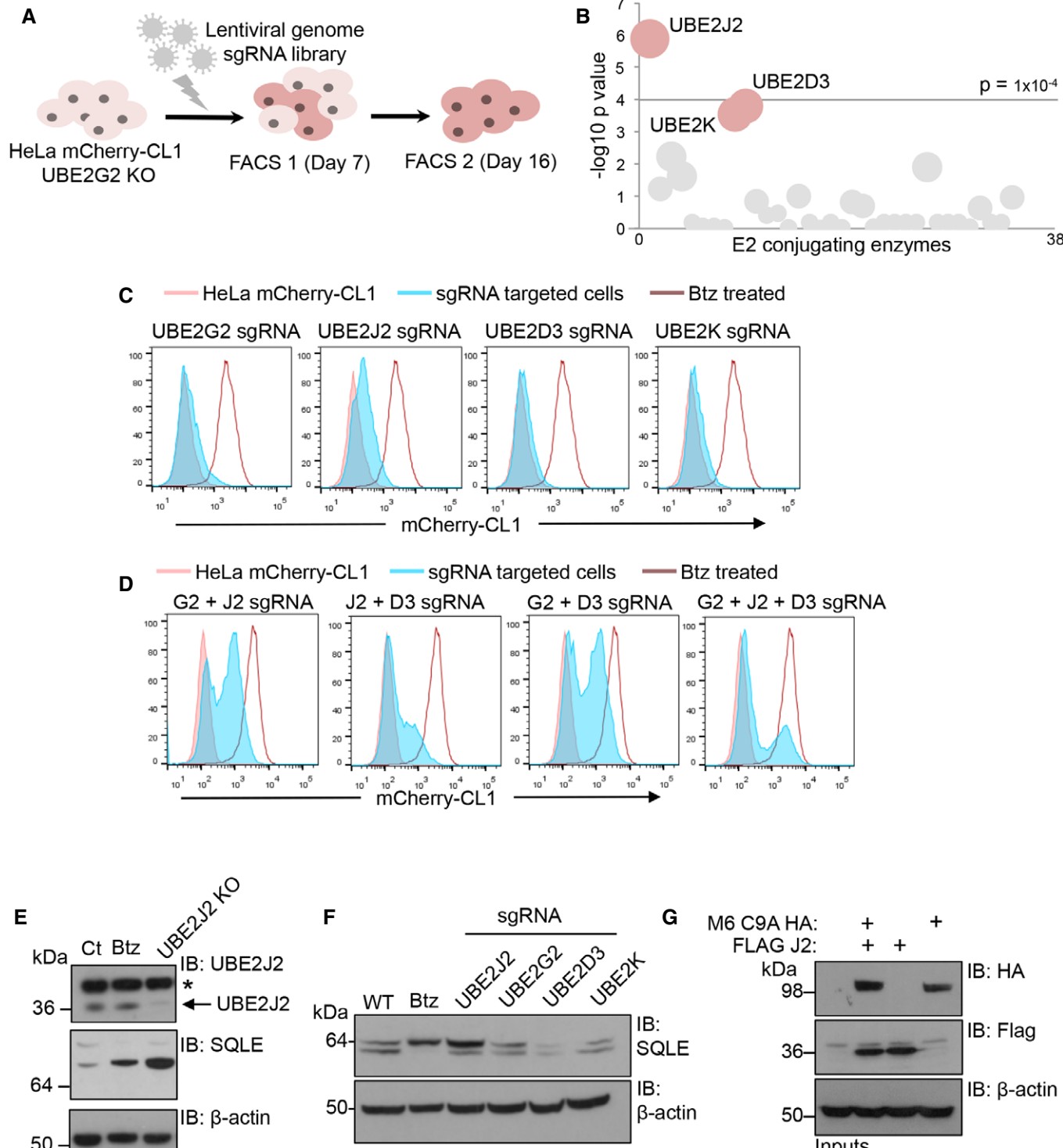

**Figure 6.**

**Figure 6.  A CRISPR/Cas9 forward genetic screen identifies Ube2J2 as the E2 conjugating enzyme for MARCH6.**

A    Schematic of the genome-wide CRISPR screen in HeLa mCherry-CL1 UBE2G2 null cells using the Brunello CRISPR knockout library.

B    Bubble plot showing the E2 conjugating enzymes identified in the screen. Bubbles represent the genes enriched in the mCherry[HIGH] population compared to the unsorted library population (significance threshold > −log(p)4). The size of each bubble is proportional to the number of active sgRNAs identified. Data were analysed using the MAGeCK algorithm.

C, D   Validation of E2 conjugating enzymes identified in the screen. HeLa mCherry-CL1 cells were transiently transfected with sgRNA and Cas9 targeting UBE2G2, UBE2J2, UBE2D3, UBE2K singly (C) or in combination (D) and analysed 7 or 10 days post-transfection by flow cytometry. 20 nM bortezomib 16 h was used as a control for mCherry-CL1 stabilisation.

E    HeLa mCherry-CL1 cells were transduced with Cas9 and UBE2J2 sgRNA, sorted by FACS to enrich for mCherry[HIGH] cells and analysed by immunoblot for SQLE levels. UBE2J2 depletion was also confirmed. *Non-specific band.

F    Immunoblot for SQLE levels in HeLa cells targeted with sgRNA to UBE2J2, UBE3G2, UBE2D3 or UBE2K to generate mixed KO populations. 20 nM bortezomib 16 h was used as a control for SQLE stabilisation.

G    Representative HA co-immunoprecipitation using HeLa cells stably expressing MARCH6 (M6) C9A-HA with or without stable expression of FLAG-UBE2J2 (J2). Cells were lysed in 1% digitonin and immunoprecipitated with an HA antibody. The association with FLAG-UBE2J2 was determined after detergent washes with 1% Triton X-100.

HO-1 degradation [27]. However, siRNA-mediated depletion of TRC8 alone was never sufficient to rescue endogenous HO-1 degradation [27], suggesting that a further E3 ligase was involved. We also noted that incorrect membrane insertion or intramembrane cleavage of HO-1 by SPP would leave a partially embedded membrane fragment, potentially resembling the membrane association of the CL1 amphipathic helix (Fig 7B). As both MARCH6 and TRC8 facilitated the degradation of the CL1-degron, we asked whether MARCH6 provided the additional E3 ligase activity for the endogenous quality control of HO-1 following SPP cleavage.

We first examined whether MARCH6 associated with endogenous SPP similar to TRC8 [27] and observed that SPP was readily detected following immunoprecipitation of HA-MARCH6 or the catalytically inactive MARCH6 C9A mutant (HA-MARCH6 C9A; Appendix Fig S6A and B). We then determined whether both MARCH6 and TRC8 were involved in the SPP-mediated degradation of endogenous HO-1 in HeLa mCherry-CL1 cells by [35S]methionine/cysteine metabolic labelling and pulse-chase analysis. Endogenous HO-1 degradation is relatively slow [27], but readily visible after a 24-h chase (Fig 7C, Appendix Fig S6C), and inhibited by addition of (Z-LL)$_2$ ketone, a specific SPP inhibitor (Fig 7C, Appendix Fig S6C). Loss of both MARCH6 and TRC8 stabilised HO-1 (Fig 7C), but depletion of neither ligase alone affected HO-1 degradation (Appendix Fig S6C). To confirm the change in HO-1 levels following depletion of both ligases was not due to the presence of mCherry-CL1, we repeated the pulse-chase in HeLa cells without the

fluorescent reporter, by generating a TRC8 KO HeLa clone (Appendix Fig S6D) and subsequently depleting MARCH6 using sgRNA (Appendix Fig S6E). Again, HO-1 was only efficiently stabilised in the combined MARCH6- and TRC8-deficient cells (Fig 7D). Furthermore, we observed that depletion of both ligases altered the steady-state levels of another SPP substrate, the tail-anchored protein RAMP4 [27,46]. Endogenous levels of RAMP4 showed a small increase in TRC8-deficient HeLa cells, a further increase following MARCH6 depletion, and were only fully stabilised to the level of SPP inhibition in the combined MARCH6-/TRC8-deficient cells (Appendix Fig S6F–H).

Signal peptide peptidase intramembrane proteolysis precedes the ubiquitination and degradation of HO-1 [27]. Consistent with this finding, we noted the presence of the faster migrating cleaved form of HO-1 species in the combined MARCH6/TRC8 null cells (Fig 7C and D). To visualise this cleaved form more clearly, we overexpressed SPP in the wild-type or ligase-deficient cells (Appendix Fig S6I) and used [35S]methionine/cysteine-radiolabelling and pulse-chase analysis to follow the proteolysis of HO-1 (Fig 7E). Overexpressed SPP increased HO-1 degradation in HeLa cells as expected (Fig 7E). However, the most striking observation was the stabilised cleaved form of HO-1 in the combined MARCH6/TRC8 null cells following SPP overexpression (Fig 7E). Thus, SPP intramembrane proteolysis precedes the TRC8-/MARCH6-mediated degradation of HO-1. Together, these experiments support a combined role of MARCH6 together with TRC8

**Figure 7.  MARCH6 and TRC8 are required for the degradation of selected ER proteins following SPP-mediated intramembrane proteolysis.**

A    Volcano plot showing TMT mass spectrometry analysis of whole-cell proteome of mCherry-CL1 cells compared to combined MARCH6/TRC8 null cells. Significance is shown in the y-axis (log(2) q-value) and log(2)-fold change on the x-axis. Proteins that were highly enriched in the combined MARCH6/TRC8 null cells and not in the single MARCH6 or TRC8 KO clones (detailed in Dataset EV2) are shown in purple.

B    Schematic of membrane topology of the tail-anchored protein, HO-1, that fails to insert correctly, or before and after intramembrane cleavage by SPP, in comparison with the amphipathic mCherry-CL1 degron.

C    [35S]methionine/cysteine-radiolabelling of HO-1 in HeLa mCherry-CL1 cells (WT) or the combined MARCH6/TRC8 null cells. Cells were pulse-radiolabelled for 10 min and immunoprecipitated for HO-1 from detergent lysates at the times indicated. Cells were treated with or without the SPP inhibitor (SPP inb), 10 μM (Z-LL)$_2$ ketone for 24 h. Full-length (black) and SPP-cleaved form of HO-1 (blue) are indicated. The asterisk represents non-specific band observed in the HeLa mCherry-CL1 cells (see also Appendix Fig S5).

D    [35S]methionine/cysteine-radiolabelling of HO-1 in wild-type HeLa cells, in HeLa TRC8 KO cells (see Appendix Fig S5C and D) or in the TRC8 KO clone depleted for MARCH6 by sgRNA (M6 KO), as described. Full-length and SPP-cleaved form of HO-1 are indicated.

E    [35S]methionine/cysteine-radiolabelling of HO-1 in HeLa cells (WT) or combined MARCH6/TRC8 null cells, with or without overexpressed SPP. The SPP-cleaved form of HO-1 is indicated.

F    Schematic for mechanism of HO-1 quality control at the ER membrane, requiring both MARCH6 and TRC8 to ubiquitinate misinserted or cleaved HO-1, targeting HO-1 for proteasome-mediated degradation.

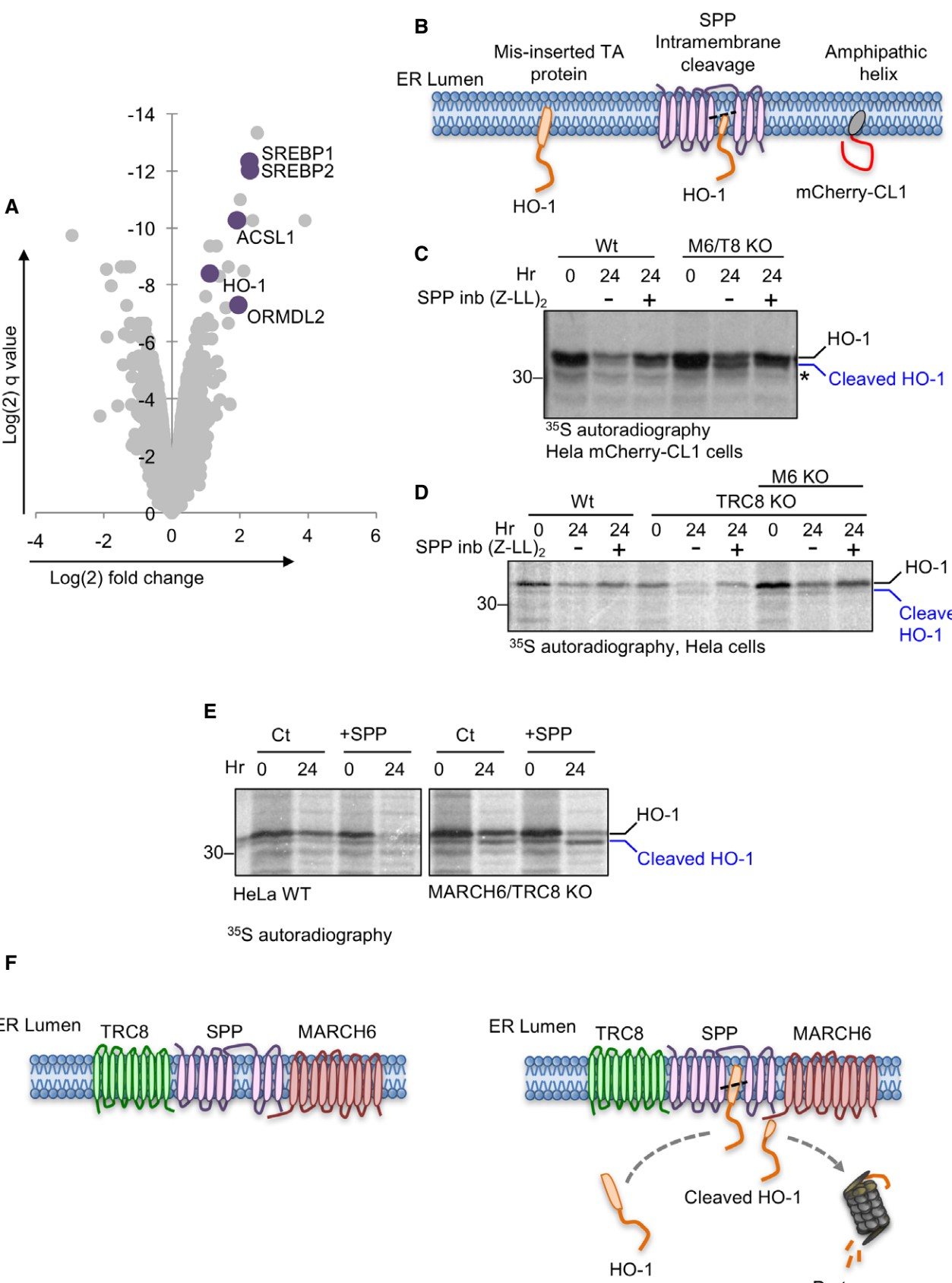

**Figure 7.**

in the ER quality control of HO-1 and other tail-anchored proteins (Fig 7F).

## Discussion

By applying forward genetic screens in human cells, we show that a soluble, hydrophobic fluorescent reporter is routed for proteasome-mediated degradation by two ER-resident ubiquitin E3 ligases, TRC8 and MARCH6. In trying to identify a physiological correlate of this amphipathic reporter protein, we find that both TRC8 and MARCH6 are involved in the degradation of proteins that have undergone SPP-mediated intramembrane proteolysis, including the HO-1 and RAMP4 tail-anchored proteins.

### Why does degradation of mCherry-CL1 occur at the ER membrane?

The simplest explanation for the membrane association of the mCherry-CL1 reporter is that the CL1 hydrophobic region is recognised by chaperones involved in targeting transmembrane proteins to the ER. Bag6, which functions as the main chaperone to promote proteasome-mediated degradation of newly transcribed, misfolded, tail-anchored proteins [47,48], did not affect mCherry-CL1 stability (Appendix Fig S5), arguing against a major role of Bag6 in the TRC8/MARCH6-mediated degradation of mCherry-CL1. In yeast, the Hsp70 and Hsp40 orthologues, Ssa1p and Ydj1p, have been implicated in CL1 degradation, with stabilisation of an Ura3p-CL1 fusion protein using the temperature-sensitive Ssa1-45 mutant [23]. However, as the other three Ssa proteins are also deleted in the Ssa1-45 strain [49,50], it is possible that stabilisation of the CL1 degron will only occur when all Ssa proteins are deleted. Given the large number of mammalian Hsp70 and Hsp40 orthologues and the likely redundancy that exists between them, it is perhaps not surprising that no chaperones were identified in our human genetic screens, although it remains likely that they are involved. An alternative explanation for the membrane association of mCherry-CL1 is that cytosolic terminally folded mCherry-CL1 associates with the ER membrane in a stochastic manner. Directly testing this model will require the *in vitro* reconstitution of mCherry-CL1 membrane association.

### Why does the degradation of mCherry-CL1 and selected ER proteins utilise two ER E3 ligases?

The involvement of two ER-resident E3 ligases to degrade a soluble protein was unexpected, as in yeast, the MARCH6 orthologue Doa10p is sufficient to degrade CL1-tagged proteins [9,23]. MARCH6 and TRC8 were both found to degrade selected substrates at the ER membrane, but differed in their ability to recognise exposed hydrophobic regions typical of misfolded proteins. Mutations altering the hydrophobicity of the CL1 degron imply that MARCH6 may detect a longer hydrophobic region than TRC8, as only the unmodified CL1 sequence was dependent on both ligases (Fig 4A and B). However, it is also possible that there may be additional factors involved in substrate recognition, and it will be of future interest to directly test our model in an *in vitro* reconstituted system.

Our findings are distinct from CL1 degradation in yeast, where mutation of any single bulky hydrophobic residue is sufficient to

stabilise the protein [51]. These differences may relate to the increased complexity of membrane insertion and degradation in more diverse organisms and the subsequent requirement for multiple ER ligases to recognise misfolded proteins or those that fail to insert correctly into membranes. Indeed, it is noteworthy that TRC8 requires short hydrophobic regions in the case of cleaved HO-1 or the CL1 degron, but can also degrade a number of proteins with different transmembrane topologies when associated with the US2 viral gene product [52]. As ER-resident E3 ligase have recently been shown to form channels for retrotranslocation of luminal proteins to the cytosol [53], it will be of future interest to explore the structural basis for the requirement of hydrophobic membrane regions for TRC8- or MARCH6-mediated degradation.

Prior studies have reported cooperative degradation of substrates by two E3 ligases through the recognition of different misfolded domains in transmembrane proteins [54] or through sequential ubiquitin modifications [55,56]. These mechanisms do not explain the involvement of both MARCH6 and TRC8 in the degradation of mCherry-CL1 or HO-1, as the ligases seem to recognise the substrates independently of each other. Alternatively, endogenous TRC8 and MARCH6 expression may be rate-limiting for degradation. Indeed, TRC8 or MARCH6 overexpression reduced the basal levels of mCherry-CL1 fluorescence, and TRC8 overexpression completely restored mCherry-CL1 degradation in the combined MARCH6/TRC8 null cells (Fig 3D). Although ectopic MARCH6 or TRC8 may alter substrate specificity, these findings are compatible with the ligases being rate-limiting for degradation at endogenous levels. However, it is noteworthy that only ectopic TRC8 completely restored the degradation of mCherry-CL1 in combined MARCH6-/TRC8-deficient cells, which may relate to prior observations that MARCH6 is rapidly autoubiquitinated and degraded [42], whereas TRC8 is more stable when overexpressed.

The identification of overlapping roles for TRC8 and MARCH may explain seemingly contradictory reports regarding TRC8 and MARCH6 substrates. For example, TRC8 was initially implicated in the regulation of SREBP-1/2 [57,58], but loss of MARCH6 was recently shown to increase SREBP-1/2 signalling [59]. Our identification of SREBP-1/2 (Fig 6A) as a potential MARCH6/TRC8 substrate may reconcile these findings. However, the TMT mass spectrometry analysis also shows that both ligases function completely independently of each other and target distinct substrates (Fig 6A and Dataset EV3). This is supported by our finding that TRC8 has no involvement in the MARCH6-mediated degradation of SQLE (Fig 2D) and by prior work showing a role for just TRC8 in US2-mediated MHC class I ubiquitination and degradation [19].

Therefore, what is in common between the CL-1 degron and HO-1 that allows recognition by both MARCH6 and TRC8? The answer is not yet clear, but given the preference of TRC8 and MARCH6 for certain hydrophobic regions, the likely explanation relates to the partial association/insertion of the substrate in the ER membrane. Indeed, Guna *et al* [60] now elegantly show that at least two pathways exist for membrane insertion of tail-anchored proteins, dependent on the hydrophobicity of the membrane spanning region, and it is therefore possible that MARCH6 and TRC8 recognise discrete pools of misfolded proteins dependent on their insertion pathway or hydrophobic regions. Alternatively, the recognition of soluble hydrophobic substrates at the ER membrane may be dependent on

the strength of their membrane association or dynamic exchange with a soluble substrate pool.

## MARCH6 and TRC8 associate with SPP for the physiological turnover of endogenous tail-anchored proteins

The observation that TRC8 [27,61] and now MARCH6 associate with SPP highlights the importance of intramembrane proteolysis in the ER protein quality control. While the association of TRC8 with SPP has been confirmed in several studies, the requirement for TRC8 was based on overexpression of the dominant negative TRC8 ΔRING mutant [27,61], and TRC8 depletion was insufficient to prevent HO-1 degradation. These data predicted involvement of an additional E3 ligase, which we now show to be MARCH6. Our data suggest that SPP proteolysis of HO-1 precedes recognition by the E3 ligases, as the cleaved form of HO-1 is readily detected in the MARCH6/TRC8 double KO cells, particularly following SPP overexpression (Fig 7E). This finding is consistent with our prior study, where only the SPP-cleaved form of HO-1 accumulated following proteasome inhibition [27]. Furthermore, these observations complement recent reports in yeast where the MARCH6 orthologue Doa10p facilitates the proteasome-mediated degradation of the zinc transporter Zrt1p following intramembrane proteolysis by the SPP-like protein Ypf1p [62].

The majority of ERAD substrates are overexpressed misfolded proteins where the rate of protein turnover is rapid. Assessing the turnover of endogenous proteins has proved challenging, mainly due to the comparatively rare abundance of the misfolded substrate, but where this has been examined the rate of turnover is still faster than for SPP-mediated HO-1 degradation. The endogenous HRD1 substrate, core-glycosylated CD147, has a half-life around 90 min [63], whereas endogenous HO-1 is slowly degraded (Fig 7C–E) with a half-life of 6–8 h [27]. SPP-mediated degradation may therefore be distinct from other forms of ERAD or better reflects the normal turnover of an endogenous substrate. Whether this slower rate of protein turnover occurs with other endogenous ER proteins and whether MARCH6 and TRC8 are both involved in the SPP-mediated degradation of other endogenous substrates warrant further investigation. However, our findings highlight the roles of MARCH6 and TRC8 in protein quality control pathways.

## Materials and Methods

### Cell culture, antibodies and reagents

HeLa and HEK 293ET cells were maintained in Dulbecco's modified Eagle's medium (DMEM; Gibco) supplemented with 10% FCS. KBM7 cells were maintained in Iscove's modified Dulbecco's medium (IMDM; Gibco) supplemented with 10% foetal calf serum (FCS), GlutaMax and 1% penicillin/streptomycin. HeLa, HEK293ET and KBM7 cells were authenticated by STR profiling (Eurofins Genomics).

The following antibodies were used: mCherry (rabbit, ab183628; Abcam), AUP1 (rabbit, HPA007674; Atlas Antibodies), β-actin (mouse, A2228; Sigma-Aldrich), calnexin (rabbit, ab75801; Abcam), GFP (mouse, 11814460001; Roche), GAPDH (mouse, GTX627408; GeneTex), HA (mouse, 16B12; Covance), HO-1 (mouse, ab13248;

Abcam, for immunoblotting), HO-1 (rabbit, A303-662A; Bethyl, for immunoprecipitation), RAMP4 (rabbit, sc-85114; Santa Cruz), signal peptide peptidase (rabbit, ab190253; Abcam), squalene monooxygenase (rabbit, 12544-1-AP; Proteintech), tubulin (mouse, 14-4502; eBioscience), UBE2G2 (mouse, sc-100613; Santa Cruz), TRC8 (rabbit, H-89; Santa Cruz Biotechnology), gp78 (rabbit, 18675; Proteintech), UBE2J2 (rabbit, 17713-1-AP; Proteintech) and ubiquitin (VU-1; LifeSensors). KDEL (rat polyclonal) antibody was kind gift from Geoff Butcher. Bag6 (rabbit polyclonal) and Ubl4A (rabbit polyclonal) antibodies were kind gifts from Manu Hegde.

Reagents used: bortezomib (Velcade, gift from Alfred Goldberg); (Z-LL)2ketone (421050; Calbiochem); puromycin, hygromycin and blasticidin (Cambridge Bioscience (10 μg/ml)); ProLong® Gold Antifade Reagent with DAPI (8961, Cell signaling technology®).

### Plasmids

Lentiviral plasmids used the pHRSIN backbone [64] and the packaging plasmids pMD.G (lentiviral VSVG) and pMD.GagPol (lentiviral Gag/Pol). IMAGE cDNA clones for AUP1 (IMAGE ID: 5578910) and MARCH6 (IMAGE ID: 40148713) (Source Bioscience) were cloned into pHRSIN.pSFFV.pGK.Puro with an N-terminal or C-terminal HA tag (HA-AUP1 and MARCH6-HA). pDEST17-UbE2J2 was a gift from Wade Harper (Addgene plasmid #15794) and cloned using Gibson Assembly® to create pHRSIN-pSFFV-FLAG-UBE2J2-pGK-Blasto. pMXs-3XHA-EGFP-OMP25 was a gift from David Sabatini (Addgene plasmid # 83356). pHRSIN.pGK.Puro HA-UBE2G2 was cloned from pcDNA3 Ube2g2-myc (gift from John Christianson). pHRSIN.pGK.-Puro TRC8-HA and pHRSIN.pGK.Puro TRC8ΔRING-HA were cloned from pcDNA6 TRC8-HA [19]. The catalytically inactive mutants (MARCH6 C9A and UBE2G2 C89A) were generated using site-directed mutagenesis. The GFP-CL1 (GFPu) construct was a gift from Ron Kopito. The CL1 sequence was amplified and cloned at C-terminus of mCherry (pHRSIN-mCherry PGK-Puro). CL1 mutations were introduced by oligos (Sigma) and cloned into pHRSIN-mCherry lentivirus vector. The SPP-expressing lentiviral plasmids (pHRSIN SPP-myc) and the ubiquitin lentiviral constructs, expressing a His-ubiquitin/ GFP (pHRSIN His-Ub-GFP), which result in co-translational cleavage of GFP from ubiquitin, have been described previously [27,38].

### Lentiviral production and transduction

Lentivirus was prepared by transfection using Mirus Trans-IT®-293 Transfection Reagent in HEK293ET cells as previously described [29,33]. The viral supernatant was collected after 48 h and filtered (0.45-μm filter), and cells were transduced by spin infection at 750 × *g*, 37°C for 1 h.

### KBM7 forward genetic screen

The forward genetic screen in near-haploid KBM7 cells was performed as previously described [29,30]. Briefly, $1 \times 10^8$ clonal KBM7 cells expressing the mCherry-CL1 reporter were transduced with the retroviral Z-loxP-GFP gene-trap supernatant containing 10 μg/ml hexadimethrine bromide (Polybrene) at 750 × *g* (37°C) for 1 h. Transduction efficiency was measured by flow cytometry after 72 h. After 8 days, cells were sorted for the mCherry^HIGH population by FACS. A second round of FACS was performed 10 days later for

mCherry[HIGH] cells. DNA was then extracted from these mCherry[HIGH] cells and a library of gene-trap retrovirally transduced clonal KBM7 cells expressing the mCherry-CL1 reporter, which had been subjected to FACS. Gene-trap integration sites were identified by Illumina MiSeq as previously described [29,30]. For the bubble plot, the degree of enrichment in the selected population compared to the unsorted cells was calculated using a Bonferroni-corrected one-sided Fisher's exact test.

### CRISPR/Cas9 forward genetic screen

HeLa mCherry-CL1 UBE2G2 null cells stably expressing Cas9 were transduced with the CRISPR knockout Brunello library [43] at an MOI of ~30%. Uninfected cells were removed by puromycin selection. After 7 days, mCherry[HIGH] cells were enriched by FACS, similar to the KMB7 screen. Cells were cultured for a further 9 days before a second round of FACS to select for the mCherry[HIGH] cells. DNA from the unsorted library and sorted cells was extracted using the Puregene® Core Kit A (Qiagen) according to the manufacturer's instructions, and DNA was amplified using two rounds of PCR as previously described [32]. DNA was sequenced by Illumina HiSeq as described [32], and analysed using the MAGeCK algorithm [65].

SgRNA amplification primers (5′–3′):
Outer_Fwd GCTTACCGTAACTTGAAAGTATTTCG,
Outer_Rev GTCTGTTGCTATTATGTCTACTATTCTTTCC,
P5_inner_Fwd  AATGATACGGCGACCACCGAGATCTACACTCTCTTGTGGAAAGGACGAAACACCG
P7_index_inner_Rev  CAAGCAGAAGACGGCATACGAGAnnnnnnnnGTGACTGGAGTTCAGACGTGTGCTCTTCCGATCTTCTACTATTCTTTCCCCTGCACTGT
Illumina sequencing primer: ACACTCTCTTGTGGAAAGGACGAAACACCG

### Flow cytometry

Cells were harvested, washed in PBS before fixing in 1% paraformaldehyde, PBS (20 min), and run on a BD LSRFortessa (BD Biosciences). FlowJo was used to analyse the data.

### CRISPR-Cas9 targeted deletions

Gene-specific CRISPR sgRNA oligonucleotide sequences were selected using the GeCKO v2 library, except for MARCH6 sgRNA1 and 4, which were designed using the Broad Institute design algorithm (http://portals.broadinstitute.org/gpp/public/analysis-tools/sgrna-design). Sense and antisense sgRNAs oligonucleotides were designed with 5′ CACC and 3′ CAAA overhangs, respectively. The sgRNAs were cloned into either the pSpCas9(BB)-T2A vector for transient transfection [66]. SgRNA were cloned into LentiCRISPRv2 or pKLV-U6gRNA(BbsI)PGKpuro2A-BFP for lentivirus production [67,68]. Null clones were isolated by serial dilution of the sgRNA targeted populations, and KO clones identified by immunoblot for the targeted protein. As there was not suitable antibody for MARCH6, null clones were validated by identifying an indel and by protein levels of a known target (SQLE). For experiments using mixed KO populations, cell was typically lysed 7–14 days post-transfection or transduction and analysed by flow cytometry or immunoblot. The following sgRNA were used:

MARCH6 sgRNA1 CCCCACCGTTCAATGCTGCG
MARCH6 sgRNA3 CGCCGACTTACCTTCCTCCG
MARCH6 sgRNA4 TATCATCCTTGTGTATGTAC
UBE2G2 sgRNA1 CATGGGCTACGAGAGCAGCG
UBE2G2 sgRNA2 GTTGGGATGAAACATCTCAC
TRC8 GCACGATGCAGAACCGGCTT
AUP1 GAGCCCTAGCACCGCACACA
Gp78 sgRNA1 GTTAGCTGGTCCGGCTCGCC
Gp78 sgRNA2 CGGCGAGCCGGACCAGCTAA
UBE2J2 AGAATCCTTACCTTCATAAG
UBE2D3 GAATGACAGCCCATATCAAGG
UBE2K GCAATGACAATAATACCGTG

### Cell lysis and immunoblotting

Detergent cell lysates were prepared in digitonin buffer (1% digitonin, 50 mM Tris pH 7.4, 100 mM NaCl, 1 mM PMSF, protease infibitors), Triton X-100 buffer (1% Triton X-100, 100 mM NaCl, 50 mM HEPES pH 7.4, 1 mM PMSF, protease inhibitors) or RIPA buffer (50 mM Tris pH 8.0, 150 mM NaCl, 0.1% SDS, 1% NP40, 0.5% deoxycholate, protease inhibitors). Cells were lysed for 30 min on ice, and lysates were subjected to centrifugation at $16,900 \times g$ for 10 min. Sample buffer was added to supernatant, and samples were heated at 75°C for 10 min, except for immunoblots for TRC8 and gp78, which were heated at 50°C for 20 min, and for MARCH6, which were heated at 37°C for 30 min.

Proteins were resolved on SDS–PAGE and transferred to PVDF membrane (Immobilon-P). Membranes were blocked in 5% (w/v) skimmed milk powder in PBS containing 0.2% Tween 20 for 1 h at room temperature. Proteins were separated by SDS–PAGE, transferred to methanol activated Immobilon®-P 0.45 µm PVDF membrane, probed with appropriate primary and secondary antibodies and developed using SuperSignal™ West Pico or Dura Chemiluminescent Substrates (Thermo Scientific).

### Immunoprecipitation

Digitonin or Triton X-100 lysates were "precleared" by incubation with Sepharose CL4B/Protein A for 1 h at 4°C. The supernatants were then incubated with Protein A and appropriate antibodies, or 10 µl EZviewTM Red Anti-HA beads (Sigma-Aldrich) for 3 h at 4°C. Resins were washed in 1% Triton X-100, and the bound proteins were eluted in the sample buffer and separated by SDS–PAGE prior to immunoblotting. For mCherry-CL1 ubiquitination, immunoprecipitation was performed as described, with the addition of 10 mM IAA (iodoacetamide) and 10 mM NEM (N-ethylmaleimide) to the lysis buffer.

### Cell fractionation

Cells were resuspended in break buffer (20 mM HEPES pH 7.4, 0.5 mM MgCl$_2$, 0.13 M sucrose, 50 mM NaCl, 1 mM PMSF, phosphatase inhibitors cocktail 2 (Sigma) and protease inhibitor mixture (Roche)) and passed through a ball-bearer homogeniser 20 times using 0.8 µm ball. An aliquot of the sample was reserved as the "total lysate", and the remainder was centrifuged at $500 \times g$ for 10 min at 4°C to pellet the nuclei and cell debris. The supernatant was ultracentrifuged at $150,000 \times g$ for 1 h at 4°C (Beckman

TLA-55) to obtain the cytosol and membrane fractions. The membranes were washed twice with break buffer. The membrane fraction was resuspended in break buffer in equivalent volume to cytosol fraction. Sample buffer was added to each fraction, and analysis was performed by immunoblot.

### [$^{35}$S]methionine/cysteine-radiolabelling and immunoprecipitation

Pulse-chase radiolabelling experiments were performed as described previously with some modifications [27]. Wild-type or mCherry-CL1 HeLa cells ($4 \times 10^6$) were washed and starved in 1 ml of methionine-/cysteine-free medium (R7513, Sigma) for 30 min at 37°C. Cells were labelled with 4 MBq of EasyTaq express $^{35}$S-Promix (Amersham Life Sciences/Perkin Elmer) in 100 µl of starve medium for 10 min at 37°C. 5 ml of chase medium (10% FCS, DMEM) was added to the cells, and 1 ml was immediately removed (pulse sample) and washed in 20 ml PBS. The cells were chased at 37°C for indicated times (1 ml was removed per chase sample). Cells were then pelleted and lysed at 4°C in RIPA buffer. After centrifugation ($10,000 \times g$, 4°C, 10 min), the supernatants were precleared for 1 h at 4°C, and IP were performed with rabbit anti-mCherry antibodies. Proteins were resolved by SDS–PAGE and detected by autoradiography.

[$^{35}$S]methionine/cysteine-radiolabelling for HO-1 was performed as described with some modifications. After initial labelling, 3 ml of chase media were added to the cells and 1 ml removed as the pulse sample (stored in −20°C). The cells for the chase were washed three times and plated in 6-well plates in chase medium, with or without 10 µM (Z-LL)$_2$ ketone, and incubated for 24 h at 37°C. Cells were then washed with PBS and lysed at 4°C in RIPA buffer. After centrifugation ($10,000 \times g$, 4°C, 10 min), the supernatants were precleared as described, and HO-1 (rabbit polyclonal) immunoprecipitated. Proteins were resolved by SDS–PAGE and detected by autoradiography or phosphorimager.

### Immunofluorescence

Cells were cultured on glass coverslips, washed in PBS and fixed in 4% (w/v) PFA, PBS at room temperature for 30 min. Cells were then permeabilised by incubation for 15 min at room temperature in 0.1% Triton X-100/PBS. Non-specific sites were blocked by incubation with 1% BSA in PBS for 1 h at room temperature. Coverslips were incubated with primary antibody for 1 h and washed in PBS and fluorophore-conjugated secondary antibodies (Alexa-488, Alexa-567 or Alexa-647, Molecular Probes) applied for 30 min. Coverslips were mounted to microscope slides using ProLong® Gold Antifade with DAPI. Imaging was performed on a Zeiss LSM880 confocal microscope.

### Fluorescent PCR and capillary electrophoresis

Putative MARCH6 null clones were screened for the presence of frameshift-inducing small insertions or deletions (indels) using previously described techniques [69]. Briefly, genomic DNA was extracted from clones using Gentra Puregene Core Kit A (Qiagen) and genomic loci of interest amplified in a PCR with Phusion polymerase (New England Biolaboratories) and primer pairs containing fluorescent modifications on the 5′ end of reverse primers.

Amplification was initially assayed using gel electrophoresis. The samples were then prepared for capillary electrophoresis in a 96-well plate format, where 2 µl of diluted PCR product was mixed with 9.8 µl Hi-Di Formamide (Life Technologies) and 0.2 µl GeneScan 500 LIZ Size Standard (Life Technologies). Capillary electrophoresis was performed on a 3730xL Genetic Analyser (Applied Biosystems) using a genotyping protocol. Results were analysed by Gene Mapper 5 (Applied Biosystems). Primer sequences were as follows: MARCH6 sgRNA1 Fwd: 5′-TTGTTGGCAGATTGTTTGCAGG-3′, MARCH6 sgRNA1 Rev: 5′-[6FAM]TCTCTCTTCATCTTCACCCAAACC-3′, MARCH6 sgRNA3 Fwd: 5′-AGAGTGTGTCGGTCAGAAGGA-3′, MARCH6 sgRNA3 Rev: 5′-[6FAM]TGGGTAATTACGTGTACCTGGA-3′.

### Mass spectrometry

Cells were harvested and lysed in 2% SDS/TEAB, and 50 µg of each sample was taken digested with tryspin using a modified SDC-assisted FASP protocol as previously described [70]. Samples were subsequently labelled with TMT reagents and pooled and cleaned up using SEP-PAK C18 (Waters) prior to high pH RP fractionation as previously described [71]. High pH fractions were pooled orthogonally into 24 samples for analysis by LC-MS on an Orbitrap Fusion (Thermo Fisher Scientific), utilising synchronous precursor selection mode to isolate reporter ions as previously described [71]. Data were analysed using the MASCOT (Matrix Science) search node within Proteome Discoverer v1.4 (Thermo Fisher Scientific). Quantified peptides were output to Inferno RDN [72], logged, normalised (quantile) and rolled up to protein abundance. Statistical differences between triplicate control and triplicate double knockout protein abundances were assessed using an implementation of LIMMA within the R environment, including Benjamini–Hochberg correction for multiple hypothesis testing [73]. The resulting $q$-values are reported.

### Data availability

The mass spectrometry proteomic data have been deposited to the ProteomeXchange Consortium via the PRIDE [74] partner repository with the dataset identifier PXD008955.

**Expanded View** for this article is available online.

### Acknowledgements
We thank the Nathan laboratory for their helpful discussions and Stuart Bloor for the SPP reagents and helpful discussions. We also thank Peter Bailey for his assistance with the analysis of the CRISPR genetic screen. This work was supported by a Wellcome Trust Senior Clinical Research Fellowship to JAN (102770/Z/13/Z), Wellcome Trust Principal Research Fellowship to PJL (084957/Z/08/Z) and the Medical Research Council (ASD and JAN). The Cambridge Institute for Medical Research is in receipt of a Wellcome Trust Strategic Award (100140). Dr. James Nathan is a Lister Prize Fellow.

### Author contributions
SS-B, ASD, SPB and JAN conceptualised the project and performed the investigations. SS-B, ASD, PJL and JAN wrote the manuscript. ITL performed investigations. DJHB and PJL provided resources. JCW carried out the mass spectrometry experiments, supervised by PJL. JAN was responsible for overall supervision of the project.

## Conflict of interest

The authors declare that they have no conflict of interest.

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
