## [Review Process File · EMBO Reports]

MARCH6 and TRC8 facilitate the quality control of cytosolic and tail-anchored proteins

Sandra Stefanovic-Barrett, Anna S Dickson, Stephen P Burr, James C Williamson, Ian T Lobb, Dick JH van den Boomen, Paul J Lehner, James A Nathan

Review timeline:

Submission date:	7 December 2017
Editorial Decision:	18 December 2017
Revision received:	12 January 2018
Editorial Decision:	8 February 2018
Revision received:	14 February 2018
Accepted:	15 February 2018

Transaction Report:

1st Editorial Decision

18 December 2017

Thank you for the transfer of your research manuscript to our journal and thank you again for providing a point-by-point response to the comments of the referees who have evaluated your manuscript for The EMBO Journal.

As discussed earlier, we would like to give you the opportunity to submit a revised version of your manuscript to EMBO reports. The remaining concerns of the referees can be addressed by textual changes, as outlined in the referee reports and in your point-by-point response. However, I notice the concern of referee 3 that the study, as it stands, does not provide evidence that MARCH6 or TRC8 directly bind and ubiquitylate the mCherry-CL1 reporter or HO-1. Since this represents an important point, this concern should be addressed experimentally, e.g. by monitoring the ubiquitylation status of CL-1.

Given the overall constructive comments, we would like to invite you to revise your manuscript with the understanding that the referee concerns (as detailed above and in their reports) must be fully addressed and their suggestions taken on board. Please address all referee concerns in a complete point-by-point response.

Revised manuscripts should be submitted within three months of a request for revision; they will otherwise be treated as new submissions. Please contact us if a 3-months time frame is not sufficient for the revisions so that we can discuss the revisions further.

Browsing through the manuscript myself, I noticed the following things that we need upon re-submission:

- Please provide a separate Conflict of interest paragraph after the section on Author contributions.

- Please provide a callout for Figure 3G in the main text.
- Please provide a scale bar for Fig. 1E, and a larger (better visible) scale bar for Fig. 3A.
- Please provide the Supplementary Information as a single pdf called Appendix. The Appendix includes a table of content on the first page, all figures and their legends. Please follow the nomenclature Appendix Figure S1 throughout the text and also relabel the figures according to this nomenclature.
- Please provide the Supplementary Files 1 - 3 as Expanded View datasets (Dataset EV_x) and provide the legend in the first line of the Excel files. Please note that large-scale datasets such as mass spectrometry data should be deposited in one of the relevant public databases. Please provide the accession code in a "Data Availability" section at the end of Materials & Methods.
- Please reformat the references according to the numbered style of EMBO reports 'Scientific reports'. The respective EndNote style file can be downloaded from our Guide to Authors (<https://drive.google.com/file/d/0BxFM9n2IEE5oOHM4d2xEbmpxN2c/view>)
- Finally, EMBO reports papers are accompanied online by A) a short (1-2 sentences) summary of the findings and their significance, B) 2-3 bullet points highlighting key results and C) a synopsis image that is 550x200-400 pixels large (width x height). You can either show a model or key data in the synopsis image. Please note that the size is rather small and that text needs to be readable at the final size. Please send us this information along with the revised manuscript.

I look forward to seeing a revised version of your manuscript when it is ready. Please let me know if you have questions or comments regarding the revision.

1st Revision - authors' response

12 January 2018

Referee #1:

There are considerable improvements to the manuscript and some concerns raised previously have been addressed. However, some concerns remain and several new ones have arisen. This reviewer still considers the findings interesting and a valuable contribution to the field. Therefore, I consider this work suitable for the audience of the EMBO Journal, as long as the concerns raised below are properly addressed.

We appreciate that this reviewer still considers the findings interesting and valuable. We have addressed their remaining concerns.

Major points:

In Figure 2, the individual contribution of TRC8 and MARCH6 to mCherry-CL1 degradation appear to be different among the various panels. In Figure 2B, a MARCH6 KO results in a higher mCherry-CL1 signal than TRC8 KO. This appears to contradict the pulse-chase data in Figure 2E-F, and the data in 4B. Could the authors clarify the differences observed between TRC8 and MARCH6 depletion/KO among these various figures?

The same TRC8 KO clone (clone 10) was used for the flow cytometry and pulse chase analyses in Figure 2 and Figure 5. The apparent discrepancy relates to inter-experimental variation. We initially screened several TRC8 and MARCH6 null clones, which all showed similar levels of the mCherry-CL1 (see below), in support of the pulse chase analysis in Figure 2E-F. Moreover, we have also measured the degradation of mCherry-CL1 in unrelated TRC8 and MARCH6 KO clones (clones 4 and 40) to those used in Figure 2 (clones 10 and 39 respectively), which give similar intermediate phenotypes (see below).

In this reviewer's opinion the results from figure 4B are properly interpreted. It is clear that CL1 instability is dependent on its hydrophobicity, which is expected based on previous work (for example Johnson, P. et al. 1998 Cell). However, while the data presented shows TRC8 responding robustly to hydrophobicity changes, the contribution of MARCH6 appears negligible (please see previous point).

In the mixed KO populations of MARCH6, mCherry-CL1 levels are very similar between the two experiments (Figure 2A and 4B). Differences in mCherry-CL1 levels reflect the efficiency of the sgRNA transfection. However, the key point regarding these studies is that mCherry-CL1 levels only reach that of proteasome inhibition in the combined MARCH6/TRC8 deficient cells, whereas the hydrophobic mutations are stabilised to the same level as proteasome inhibition with depletion of just TRC8.

In many places the authors rephrased the relation between TRC8 and MARCH6. However, in page 13 it is mentioned multiple times that "TRC8 and MARCH6 are both required" for CL1 degradation. This is not correct and should be rephrased as the two ligases appear to be redundant (Fig 3D and F).

This has been corrected to state that both ligases can degrade mCherry-CL1. We have also rephrased the relationship between TRC8 and MARCH6 throughout the manuscript.

Minor points:

- Page 7, second paragraph. It should read "it no longer..." instead of "it longer..."

This has been corrected.

- The authors refer to unmodified mCherry-CL1 as wildtype. However, such classification does not seem appropriate as CL1 is an artificial polypeptide resulting from the out of frame translation of a yeast gene.

We agree that the use of wildtype CL1 is ambiguous and have corrected this in the manuscript.

- In page 10 it is stated that "Given the potential similarity between the CL-1 degron and HO-1, and our prior observation that TRC8 associates and functions with SPP (Boname et al., 2014), we examined whether both MARCH6 and TRC8 are required for endogenous quality control of HO-1." The CL1 degron is an amphipathic helix while HO-1 is a tail-anchored membrane protein. These are two very distinct modes of membrane association and the potential similarity between the two is certainly not obvious. The authors should rephrase this.

We apologise for not making this sufficiently clear. The similarity refers to the SPP processed form of HO-1, which is only partially embedded within the membrane, and the membrane-association of an amphipathic helix. We have adjusted the text to reflect this point.

- The discussion point on the role of chaperones in Ura3-CL1 degradation is not correct. In yeast, there are four Ssa proteins (Ssa1-4) that are mostly redundant. Stabilization of URA3-CL1 (and most ERAD substrates) is only detected in strains expressing a SSA1 temperature sensitive allele (SSA1-45) and lacking the additional 3 SSA proteins (see Metzger et al., 2008 and Nishikawa et al. JCB 2001). Thus, the well described redundancy in cytosolic chaperones in yeast may also be present in mammalian cells and explain the fact that no chaperones were picked up in the screen with mCherry-CL1.

We thank the reviewer for directing us towards this literature. The text has been adjusted to acknowledge the redundancy described in the yeast Ssa proteins.

Referee #2:

In summary, the revised version of the manuscript by Stefanovic-Barrett et al. now includes three major additions:

.) As requested by the reviewers, the authors performed pulse-chase experiments to confirm some of the relevant results obtained by FACS analysis.

.) They also systematically analyzed the CL1 degron by site-specific mutagenesis and revealed that TRC8 and MARCH6 may target overlapping but not identical properties on a substrate. This may at least in part explain, why two different ubiquitin ligases partake in the degradation of the investigated substrates.

.) Finally, they observed an accumulation of a proteolytic HO-1 fragment in cells that lack MARCH6 and TRC8 and that overexpress SPP, which strengthens their argument that SPP acts upstream of the ubiquitin ligases.

Although these additional data strongly improved the quality of the manuscript, I am not fully convinced, whether it now fulfills all the criteria to justify publication in the EMBO Journal. As the authors state, CL1 degron fusion constructs have been investigated in yeast and there a membrane-bound ubiquitin ligase termed Doa10 was shown to be required for degradation. It is not utmost surprising that orthologues of Doa10, with MARCH6 displaying more sequence conservation to Doa10 than TRC8, are involved in the turnover of CL1 containing constructs in mammalian cells. However, the authors provide strong evidence that MARCH6 and TRC8 constitute important components for the removal of appointed tail-anchored proteins, which represents a more physiological relevant function of the ligases. Their data indicate that MARCH6 and TRC8 display slight differences in their preference for substrates but this still does not fully explain, why both ligases appear to target discrete pools of CL1-fusions or HO-1 molecules.

We thank the reviewer for the acknowledged improvement in the manuscript.

The degree of hydrophobicity provides an indication of how MARCH6 and TRC8 may recognise their substrates, and we acknowledge that other factors may be involved. We believe the best way to address these questions will be through an in vitro reconstituted system, which forms part of our ongoing work to further understand the mechanisms involved. We have adjusted the text to reflect this point, and removed any unintended over-interpretation of our findings (p12).

I am also not completely satisfied with their interpretation of the data on the functional interaction of MARCH6/TRC8 with SPP. Given that SPP acts upstream of MARCH6/TRC8 a

downregulation/knockout of these ligases should specifically impair the degradation of cleaved HO-1 and as a result the proteolytic fragment should accumulate. But this can't be seen in Figure 6D. Furthermore, after treatment with proteasomal inhibitors full-length HO-1 is enriched in such cells (Figure 6D, outmost right lane), which indicates that additional pathways aside from MARCH6 and TRC8 route full-length HO-1 to proteasomal degradation.

We think that this concern relates to an incorrect interpretation of the experiment detailed in 7D. The SPP cleaved HO-1 fragment is visible in the double knockout cells (7D, second to last lane), and we can show that it occurs after SPP cleavage, as this processed form increased when SPP is overexpressed (Figure 7E). An SPP inhibitor (Z-LL₂), not a proteasome inhibitor, was used in the experiment shown in Figure 7D. Thus, a stabilised single migrating form of HO-1 would be expected (Figure 7D, last lane). We have labelled Figure 7D more clearly to reflect these points.

The authors argue against a role of the chaperone BAG6 and the ubiquitin ligase RNF126 in CL1 or HO-1 turnover because they failed to isolate them in their screen. However, at least two studies report on the binding of BAG6 to the CL1 degron and its requirement for CL1 fusion degradation (Minami et al., JCB 2010; Tanaka et al., FEBS J. 2016). Hence, the authors should experimentally investigate any involvement of BAG6 in CL1/HO-1 turnover or at least discuss the possible contribution of such pathways to CL1 and HO-1 degradation. Should the authors address these issues at least by changes in the text I have no major objections that would argue against publication of this manuscript in the EMBO Journal.

We had previously included data investigating the role of Bag6 in CL1 degradation, where we observed no stabilisation of mCherry-CL1 following Bag6 depletion. We removed this data in the revised manuscript, as it did not seem central to the main experimental findings. However, as it is evident that its inclusion would be helpful, we have re-incorporated this data. We have also commented on the potential role of Bag6 in the discussion (p12).

Referee #3:

The revised manuscript by Stefanovic-Barrett et al. has included many new results, which do help clarify the functional relationship between the two ligases MARCH6 and TRC8 in degradation of mCherry-CL1. The manuscript is indeed improved and reads a lot more complete. However, I still have several issues that need to be addressed before publication. Mostly importantly, there is a tendency to over-interpret many results throughout the manuscript. The most outstanding problems are listed below:

We are pleased that this reviewer found the additional studies helpful. It was certainly not our intention to over-interpret our findings and we agree that it is important to present a balanced discussion of our findings. We hope that our responses to the concerns raised below now address these issues.

- The functional relations of the two ligases: Although the authors have toned down significantly, we can still see words such as "work together" (abstract), "suggested cooperation between these ligases" (page 9). In my opinion, the new data shown in Figure 4 is completely consistent with my initial suspected model, in which mCherry-CL1 is partitioned into two populations, a soluble and a membrane-associated pool. These two E3s are each responsible for dealing with one of these populations. The fact that these E3s each work with an independent E2 also suggests that they function independently of each other. I see no evidence to conclude that these enzymes can work together or cooperate with each other. Leaving these statements in the text could potentially mislead readers.*

We apologise for the ambiguous use of 'work together' or 'cooperation' throughout the manuscript, and have removed all such references to avoid mis-interpretation. The key finding is that the ligases target the same substrates but work independently of each other for efficient protein quality control. It was certainly not our intention to overstate the mechanisms uncovered in how the ligases identify their substrates.

However, we do not agree with this interpretation of the data in Figure 4, whereby one ligase targets the soluble pool and one the membrane-associated pool. We observe that membrane association is required for both TRC8- and MARCH6-mediated degradation. Once membrane association is prevented by mutation of the hydrophobic residues, the soluble pool is not targeted by either of the ligases. A further experiment to illustrate this point is included below, whereby the amino-acid CL1-sequence has been fused to the N-terminus of mCherry. This CL1-mCherry construct, which still contains the hydrophobic residues but will not be able to form a C-terminal amphipathic helix, does not associate with the membrane and is not degraded.

• *The authors should also be cautious when interpreting the rescue experiments done by overexpression of either MARCH6 or TRC8. What is the level of the expression compared to endogenous proteins? If expressed at very high levels, the specificity of these enzymes could be changed, not necessarily reflecting a rating limiting problem.*

Overexpression does result in higher levels of MARCH6 and TRC8 than the endogenous protein, and we agree that we cannot rule out that specificity may be altered. We acknowledge this point in the text (now p13). However, the most likely explanation for TRC8 overexpression degrading all of the mCherry-CL1 pool, is due to the overlapping functions of the ligases.

• *The recognition of mCherry-CL1 by E3 ligases: In several places, the authors attribute the differential E3 requirement for degradation of different CL1 degnon-bearing reporters to difference in substrate recognition. However, there is no data in this paper suggesting that these ligases are directly involved in ubiquitination of mCherry-CL1 reporter, let alone the recognition of substrates by these ligases. Both reviewer 2 and I feel that it is necessary to explore the ubiquitination pattern of mCherry-CL1 in wild type and ligase deficient cells. Surprisingly, the authors argued that this experiment is technically challenging (I have not seen a single paper in the ubiquitin ligase field that does not validate substrate ubiquitination). I think that this deficiency needs to be fixed before publication of this paper in the EMBO journal. In addition, even if the authors can show that these ligases are required for ubiquitination of mCherry-CL1, I would still suggest that they remove statement such as "Thus, while both MARCH6 and TRC8 recognise the hydrophobic membrane associated region of the CL1 degnon" from the text. It is highly possible that substrate recognition is mediated by chaperones that escape their screen due to a variety of reasons.*

We agree that it is possible that chaperones may be involved in the recognition of CL1 degnon rather than directly through the ligases and have clarified this important point in the text (p12).

We disagree with some of these comments regarding ubiquitination. Mutations in the catalytic sites of the ligases show that their ubiquitin E3 activity is required for CL-1 degradation, and endogenous ubiquitination is often difficult to detect. However, to address this further, we now include data using overexpressed ubiquitin and immunoprecipitation of mCherry-CL1 (Appendix Figure S4A). Ubiquitinated mCherry-CL1 is observed in wildtype HeLa cells treated with the proteasome inhibitor MG132, but markedly reduced in the combined MARCH6/TRC8 null cells (Appendix

Figure S4A). The residual low level of ubiquitination observed in the MARCH6/TRC8 null cells may reflect that additional pathways can be involved, although, it is noteworthy that our pulse-chase analyses show near-complete stabilisation of mCherry-CL1 in the MARCH6/TRC8 cells, so this residual ubiquitination does not seem to alter CL1 stability (discussed, page 6).

In addition to the new ubiquitination experiment, we have also explained the overexpression competition assay using lysine ubiquitin mutants in detail (page 6, Appendix Figure S4B,C), which supports a role for both MARCH6 and TRC8 in K48-polyubiquitination of mCherry-CL1.

• The precise nature of the reported degradation pathway: The authors argue that HO-1 likely represents an endogenous substrate of the pathway identified using mCherry-CL1 as the model substrate because like mCherry-CL1, they both require simultaneous knockout of MARCH6 and TRC8 to achieve maximum stabilization. However, I notice an obvious difference between the two substrates, that is for HO-1, knockout of either MARCH6 or TRC8 completely failed to even partially stabilize it. This is in contrast to mCherry-CL1. The discrepancy seems to argue against the authors' idea that these two substrates are degraded by the same mechanism. Instead, it suggests that for HO-1, the two ligases function in a 'genetically' redundant fashion. To better substantiate the authors' model, I feel that they should test whether HO-1, upon stabilization by proteasome inhibition, is also partitioned into two populations, a soluble and a membrane-associated one, which may undergo rapid and dynamic exchange whereas the membrane association of mCherry-CL1 may be more static. This could help explain the observed difference in genetic requirement for degradation. At minimum, since substrate recognition by these ligases are poorly characterized, the authors should tone down their conclusions, and discuss all possible models.

We apologise that we were not sufficiently clear regarding the differences between the two substrates. Both ligases are involved in CL-1 and HO-1 degradation but we are not suggesting the mechanism is exactly the same. Most notably, CL-1 is an artificial degron not under endogenous regulation, unlike HO-1. This may explain why we see an intermediate phenotype with mCherry-CL1 that is not readily observed with endogenous HO-1. However, we agree that all possible models should be explored and have included this in the revised discussion (p12-14).

A similar experiment to the proteasome inhibition experiment suggested was performed in our JCB 2014 studies, and only a single pool of cleaved HO-1 accumulated, arguing against this model of partitioning into two pools (discussed p14).

Minor points:

• The first point in the rebuttal letter is incomplete "Hence, the importance of this study". I am not sure what exactly the authors wanted to say here. The authors argue that the paper is important because "we now show that MARCH6 and TRC8 differ in the recognition of hydrophobic regions within their substrates, highlighting the specificity that exists in the regulation of misfolded soluble proteins at the cytosolic face of the ER". This is clearly an overstatement as I pointed out above. As I mentioned above, the new results in Figure 4 have provided significant new insights on why degradation of mCherry-CL1 requires two ligases, but not to a level at which the authors can claim that they understand the substrate specificity for these ligases.

We agree that the details of substrate specificity are not fully understood, and this work forms part of our ongoing studies. We have clarified this point in the revised manuscript (p12/13).

• Several typos need to be fixed. Page 7, "it longer associated with membranes..."; Page 12, "...is required for a longer hydrophobic region that TRC8." In summary, I appreciate the efforts the authors have put in to improve this paper, and I also believe that the paper contains useful information for readers of the EMBO journal. That being said, one thing that I am strongly against is over-interpretation of data in order to make a paper sound interesting and significant.

The typographical errors have been corrected. We appreciate that this reviewer finds that the studies will be useful to the readers of EMBO, and reiterate that it was certainly not our intention to over-interpret our experimental findings.

Thank you for your patience while we have reviewed your revised manuscript. As you will see from the reports below, the referees are now all positive about its publication in EMBO reports and have only some minor suggestions regarding textual changes. I am therefore writing with an 'accept in principle' decision, which means that I will be happy to accept your manuscript for publication once a few minor issues/corrections have been addressed, as follows.

- In the main text one callout for Figure S6I lacks the "Appendix" (last paragraph of results)
- Please change the legend in the Datasets to "Dataset EVx" instead of "Expanded View Dataset x"
- Mass spectrometry datasets should be deposited in a machine-readable format (e.g. mzML if possible) in one of the major public database, for example Pride (<http://www.ebi.ac.uk/pride/>) or PeptideAtlas(<http://www.peptideatlas.org>) in accordance with MIAPE recommendations (<http://www.psidev.info/index.php?q=node/91>).
If you submit the mass spec data please provide the access code in a separate Data availability section at the end of Materials & Methods section.
- Finally, our data editors have checked the figure legends of your manuscript for completeness. Please find their suggested changes in the attached Word document.

If all remaining corrections have been attended to, you will then receive an official decision letter from the journal accepting your manuscript for publication in the next available issue of EMBO reports. This letter will also include details of the further steps you need to take for the prompt inclusion of your manuscript in our next available issue.

Thank you for your contribution to EMBO reports.

REFEREE REPORTS

Referee #1:

The authors addressed all my concerns appropriately and I consider the work suitable for publication in EMBO Reports.

I found two minor typos:

- Page 10, paragraph 2: "widltype HeLa". It should read wild type (or even better "control HeLa", It is ironic to call wild type to a cell line derived from a cervical cancer!)
- Last sentence of discussion: ER is misplaced in the sentence.

Referee #2:

The authors have addressed all my criticisms.

Referee #3:

Stefanovic-Barrett et al. report on a genetic screen using a CL1-mCherry reporter construct that led to the identification of two membrane-bound ubiquitin ligases, MARCH6 and TRC8, involved in cytoplasmic protein quality control. They observe that the degradation of CL1-mCherry is only partially inhibited in cells lacking one of those ubiquitin ligases and that only the combined loss of both enzymes completely abolishes the turnover of the reporter. Subsequent genetic analysis allowed them to identify the cognate E2 enzymes for these ligases, UBE2G2 with its co-factor AUP1 for TRC8 and UBE2J2 for MARCH6. Thus, both ligases constitute discrete pathways that appear to target distinct pools of the CL1-mCherry fusion. Importantly, March6 as well as TRC8 interact with Signal Peptide Peptidase (SPP) and thereby contribute to the degradation of some tail-

anchored proteins. SPP appears to first cleave selected tail-anchored proteins, which primes them for the processing by the ubiquitin ligases.

This work has already passed at least one round of evaluation and the authors have successfully addressed the comments of the reviewers. The experiments are technically sound and the manuscript is very well written. In consequence, this manuscript is now of more than average quality. I am confident that this study is of interest to a broader readership and definitely recommend publication in "EMBO reports".

2nd Revision - authors' response

14 February 2018

The authors performed all minor editorial changes.

Corresponding Author Name: James Nathan

Manuscript Number: EMBOJ-2017-97528R